# AMPK regulates ESCRT-dependent microautophagy of proteasomes concomitant with proteasome storage granule assembly during glucose starvation

**Jianhui Li[1], Michal Breker[2], Morven Graham[3], Maya Schuldiner[2], Mark Hochstrasser[1,4]***

**1** Department of Molecular Biophysics and Biochemistry, Yale University, New Haven, Connecticut, United States of America, **2** Department of Molecular Genetics, Weizmann Institute of Sciences, Rehovot, Israel, **3** Center for Cellular and Molecular Imaging, School of Medicine, Yale University, New Haven, Connecticut, United States of America, **4** Department of Molecular, Cellular, and Developmental Biology, Yale University, New Haven, Connecticut, United States of America

* mark.hochstrasser@yale.edu

**Data Availability Statement:** All relevant data are within the manuscript and its Supporting Information files.

## Abstract

The ubiquitin-proteasome system regulates numerous cellular processes and is central to protein homeostasis. In proliferating yeast and many mammalian cells, proteasomes are highly enriched in the nucleus. In carbon-starved yeast, proteasomes migrate to the cytoplasm and collect in proteasome storage granules (PSGs). PSGs dissolve and proteasomes return to the nucleus within minutes of glucose refeeding. The mechanisms by which cells regulate proteasome homeostasis under these conditions remain largely unknown. Here we show that AMP-activated protein kinase (AMPK) together with endosomal sorting complexes required for transport (ESCRTs) drive a glucose starvation-dependent microautophagy pathway that preferentially sorts aberrant proteasomes into the vacuole, thereby biasing accumulation of functional proteasomes in PSGs. The proteasome core particle (CP) and regulatory particle (RP) are regulated differently. Without AMPK, the insoluble protein deposit (IPOD) serves as an alternative site that specifically sequesters CP aggregates. Our findings reveal a novel AMPK-controlled ESCRT-mediated microautophagy mechanism in the regulation of proteasome trafficking and homeostasis under carbon starvation.

## Author summary

Protein homeostasis is critical for maintaining organismal health. The cellular dysfunction caused by accumulation and aggregation of aberrant proteins or other normally short-lived proteins is associated with aging and many human diseases, including neurodegenerative disorders, diabetes, and various types of cancer. The eukaryotic ubiquitin-proteasome system regulates numerous cellular processes and through selective protein degradation helps maintain cellular protein homeostasis under normal growth conditions. However, hundreds of cellular granules or condensates are formed during carbon

**Funding:** This work was supported by NIH grant GM083050 to M.H. and the Minerva foundation with funding from the Federal German Ministry for Education and Research to M.S. M.S. is an incumbent of the Dr. Gilbert Omenn and Martha Darling Professorial Chair in Molecular Genetics. The funders had no role in study design, data collection and analysis, decision to publish, or preparation of the manuscript.

**Competing interests:** The authors have declared that no competing interests exist.

starvation in yeast cells, including proteasome storage granules (PSGs). PSGs result from a massive relocation of proteasomes from the nucleus to the cytoplasm under these conditions. However, how cells regulate proteasome homeostasis under these conditions remains largely unknown. Here, we demonstrate that AMPK (AMP-activated protein kinase), a master cellular energy regulator, drives ESCRT (endosomal sorting complexes required for transport)-dependent microautophagy of aberrant proteasomes. This allows rapid re-mobilization of functional proteasomes from PSGs upon glucose refeeding. Previous studies had identified classical macroautophagy as a means of degrading proteasomes during starvation. Our work shows that direct uptake of proteasomes into the vacuole (lysosome) by microautophagy is a major means of proteasome elimination under limiting glucose conditions.

## Introduction

The ubiquitin-proteasome system (UPS) is a conserved proteolytic system responsible for the highly selective degradation of cellular proteins. Conjugation of ubiquitin to substrates targets them to the proteasome for degradation [1, 2]. The 26S proteasome comprises a 20S core particle (CP) with a 19S regulatory particle (RP) on one or both ends of the CP [1]. In the CP, four stacked rings are assembled from different β-subunits (β1-β7) and α-subunits (α1-α7). The RP is assembled from two multisubunit subcomplexes termed the base and lid [1]. The RP is responsible for substrate binding, deubiquitylation, unfolding, and translocation [1, 3–5].

The UPS accounts for ~80–90% of cellular proteolysis under normal growth conditions, and therefore, alterations in proteasome activity have a major impact on protein homeostasis ("proteostasis") [6, 7]. For instance, the age-related decline of proteostasis can be compensated by increasing proteasome activity, while cancer cell growth can be limited by inhibiting proteasome activity [8–10]. One way to regulate the availability of proteasomes is by formation of Proteasome Storage Granules (PSGs), which are membraneless cytoplasmic structures that collect proteasomes during specific stresses. PSGs most likely serve as an adaptive mechanism to regulate proteasome activity [11, 12]. Multiple factors have been reported to be relevant to the assembly and disassembly of PSGs, including carbon starvation [13], intracellular pH [14], protein N-α-acetylation [11], Blm10 [15], and the integrity of certain proteasome lid subunits [12, 16, 17]. Nevertheless, relatively little is known about how proteasome nucleocytoplasmic trafficking and homeostasis are regulated, particularly during carbon starvation.

Autophagy is another major means of degrading cellular components, which protects cells from damaged proteins and organelles and promotes cell survival under various stresses, such as starvation, oxidative stress, protein aggregation, and ER stress. The cellular components of macroautophagy have been extensively studied in yeast and assigned roles in selective or nonselective macroautophagy or both [18]. All forms of macroautophagy share a common set of core *au*tophagy (Atg) proteins that are involved in autophagosome initiation and formation [19]. Other Atg factors are required for specific types of selective macroautophagy [18]. Atg17 is required only for nonselective macroautophagy [20].

In comparison to macroautophagy, microautophagy is poorly understood [21]. Recently, endosomal sorting complexes required for transport (ESCRT)- and clathrin-dependent microautophagy has been described in yeast undergoing diauxic shift; the substrate protein followed in this study was the transmembrane vacuolar protein Vph1 [22]. The ESCRT machinery is an ancient system for membrane remodeling and scission; in eukaryotes it includes five distinct subcomplexes: ESCRT-0, I, II, III, and the AAA ATPase Vps4 [23]. The ESCRT pathway drives

diverse cellular processes, such as multivesicular body (MVB) formation, nuclear pore complex (NPC) quality control, virus budding, viral replication-complex assembly, macroautophagy, and microautophagy [23, 24]. How the ESCRT machinery is activated and regulated in microautophagy, however, is unclear.

Proteasome homeostasis involves a balance between proteasome assembly and degradation. Normally, proteasomes are very stable entities within the cell [25], but their degradation by autophagy is induced under certain conditions. Proteasome degradation by macroautophagy ("proteaphagy") occurs in response to nitrogen starvation and has been described in yeast, plants, and mammals [26–30]. A recent study reported that PSGs protect proteasomes from autophagic degradation during carbon starvation [31], suggesting a physiological connection between PSGs and autophagy. It is unknown how the partitioning of proteasomes between assembling PSGs in the cytoplasm and proteolysis in the vacuole is regulated.

AMP-activated protein kinase (AMPK) is a highly conserved regulator of energy homeostasis in eukaryotes. The AMPK heterotrimeric complex is composed of a catalytic α subunit (called Snf1 in *S. cerevisiae*) and two regulatory subunits: a β subunit (Sip1, Sip2, or Gal83) and a γ subunit (Snf4) [32]. The AMPK pathway is activated when energy stores are depleted, which modulates the switch from fermentation to respiration in yeast [32, 33]. Moreover, AMPK coordinates a wide range of cellular processes, such as cell growth, autophagy, metabolism, cell polarity, and cytoskeletal dynamics [34].

In this study, we demonstrate that AMPK, the ESCRT machinery, and the insoluble protein deposit (IPOD) function together in the regulation of proteasome trafficking and degradation under glucose starvation. Cells utilize AMPK- and ESCRT-dependent microautophagy to clear aberrant proteasomes through vacuolar proteolysis under these conditions and thus safeguard reversible PSG assembly and the maintenance of functional proteasomes during glucose starvation. Our cytological data suggest very high levels of microautophagy can occur under these conditions. We find that proteasomes dissociate into CP and RP complexes that are regulated through distinct trafficking mechanisms during glucose starvation. Furthermore, apparently irreversible CP aggregates accumulate if AMPK is inactivated and cells are carbon starved. In such mutants, CP aggregates concentrate in the IPOD compartment. These findings reveal the convergence of multiple cellular pathways that maintain proteasome homeostasis during changes in nutrient availability.

## Results

### A high-content screen for defects in PSG dynamics identifies AMPK and ESCRT proteins

We set out to identify factors that are involved in PSG dissipation and nuclear reimport of proteasomes upon glucose refeeding. To this end, we screened a ~5,000-strain yeast library where each strain had an integrated *RPN5-GFP* allele and a different gene deletion from the yeast gene deletion library [35] created by synthetic genetic array (SGA) methodology [14, 36]. Each strain was imaged on a high-throughput fluorescence microscopy platform [37]. The screen identified 198 potential hits (S1 Table), with multiple hits from two conserved cellular machineries, AMPK and the ESCRT machinery. The hits included two subunits of the AMPK heterotrimeric complex (Snf1 and Snf4) and multiple constituents of the ESCRT pathway (ESCRT-0 [Vps27], ESCRT-II [Vps25], ESCRT-III [Did2, Vps2/Did4, Vps24], and the AAA ATPase Vps4). Notably, Snf1 and Vps24 were also identified in a previous high-content screen for PSG formation but were not pursued further [38].

To validate the candidates from the AMPK and ESCRT complexes, we added an mCherry (mC) tag at the C-terminus of three individual proteasome subunits: Pre1-mC (a CP subunit,

β4), Rpn2-mC (a base subunit), and Rpn5-mC (a lid subunit), and we then introduced the corresponding genes into yeast strains also expressing a nuclear envelope marker, Nup49-GFP, but lacking individual candidate genes. In wild-type (WT) cells under glucose starvation, all of these tagged proteasomes localized to the cytoplasm and assembled into bright, round PSGs with dark cytoplasmic backgrounds (Fig 1A). The PSGs rapidly dissolved and proteasomes were reimported into the nucleus in ~90% of cells within 15–20 min after 2% glucose addition (Fig 1B and 1E). These observations are consistent with the original report on PSG formation and dissipation in yeast cells [13].

By contrast, cells with mutations in the ESCRT machinery displayed a mixture of diffuse cytoplasmic proteasomes and proteasome granules under glucose starvation conditions (Fig 1A and S1A Fig). Upon glucose refeeding, the mutants also showed defective proteasome granule dissipation and nuclear reimport of proteasomes, as predicted from the original screen (Fig 1B and S1B Fig). These findings suggest that the ESCRT machinery is involved in PSG assembly during glucose starvation and that normal PSG assembly is a prerequisite for efficient PSG dissipation and nuclear reimport of proteasomes upon glucose restoration.

In AMPK mutant cells lacking either *SNF4* or *SNF1*, PSGs similar to those in WT cells were observed by fluorescence microscopy (Fig 1C). This was also evident from immunogold electron microscopy. PSGs of similar size (~500 nm dia.) were observed in WT and the AMPK mutant cells that had been immunogold-labeled with anti-CP antibody (Fig 1F). However, PSG disassembly and nuclear reimport of proteasomes were defective in the AMPK mutants (Fig 1D and 1E), suggesting that PSG dissipation requires AMPK pathway signaling. Collectively, these results suggest that AMPK and the ESCRT machinery are both required for reversible PSG formation during glucose starvation and refeeding but likely for different steps.

## Proteasome macroautophagy requires the ESCRT pathway but not AMPK during nitrogen starvation

Prior work showed that AMPK plays a direct role in promoting macroautophagy in mammalian cells in response to energy deprivation [39, 40], while the ESCRT machinery has roles in multiple autophagic pathways [41]. Proteasome degradation by macroautophagy is induced by nitrogen starvation in yeast cells [26–28], and a recent study reported that PSGs protect proteasomes from autophagic degradation during carbon starvation [31]. Given these interconnections, we checked whether mutations in AMPK and ESCRT factors affected proteasome macroautophagy during nitrogen starvation by using a GFP-release immunoblot assay [28]. GFP fusion proteins delivered to the vacuole by macroautophagy are hydrolyzed by vacuolar proteases, but the GFP moiety itself is more resistant to cleavage. We assessed GFP "release" in nitrogen-starved cells expressing Pre10-GFP (a CP subunit, α7), Rpn5-GFP, or Rpn2-GFP. GFP release from all three fusions was strongly inhibited in yeast strains lacking components of the ESCRT machinery, including ESCRT-0 (Vps27), ESCRT-I (Vps28 and Vp37), ESCRT-II (Vps25), ESCRT-III (Snf7), and the Vps4 ATPase, when compared to that in WT cells (Fig 2A). The one exception was the comparable amount of free GFP release from Pre10-GFP in *vps4Δ* cells. This finding suggests that CP macroautophagy is independent of Vps4, while RP macroautophagy requires the complete ESCRT pathway. This is also consistent with proteasomes disassembling into CP and RP complexes before their trafficking to the vacuole for degradation during nitrogen starvation. By contrast, a similar amount of free GFP release from Pre10-GFP, Rpn5-GFP, and Rpn2-GFP was detected in WT, *snf4Δ*, and *snf1Δ* cells (Fig 2A), indicating AMPK is not essential for macroautophagy of proteasomes during nitrogen starvation.

We also checked the subcellular localization of Pre10-GFP, Rpn5-GFP, and Rpn2-GFP in nitrogen-starved ESCRT and AMPK mutants by fluorescence microscopy. Proteasomes were

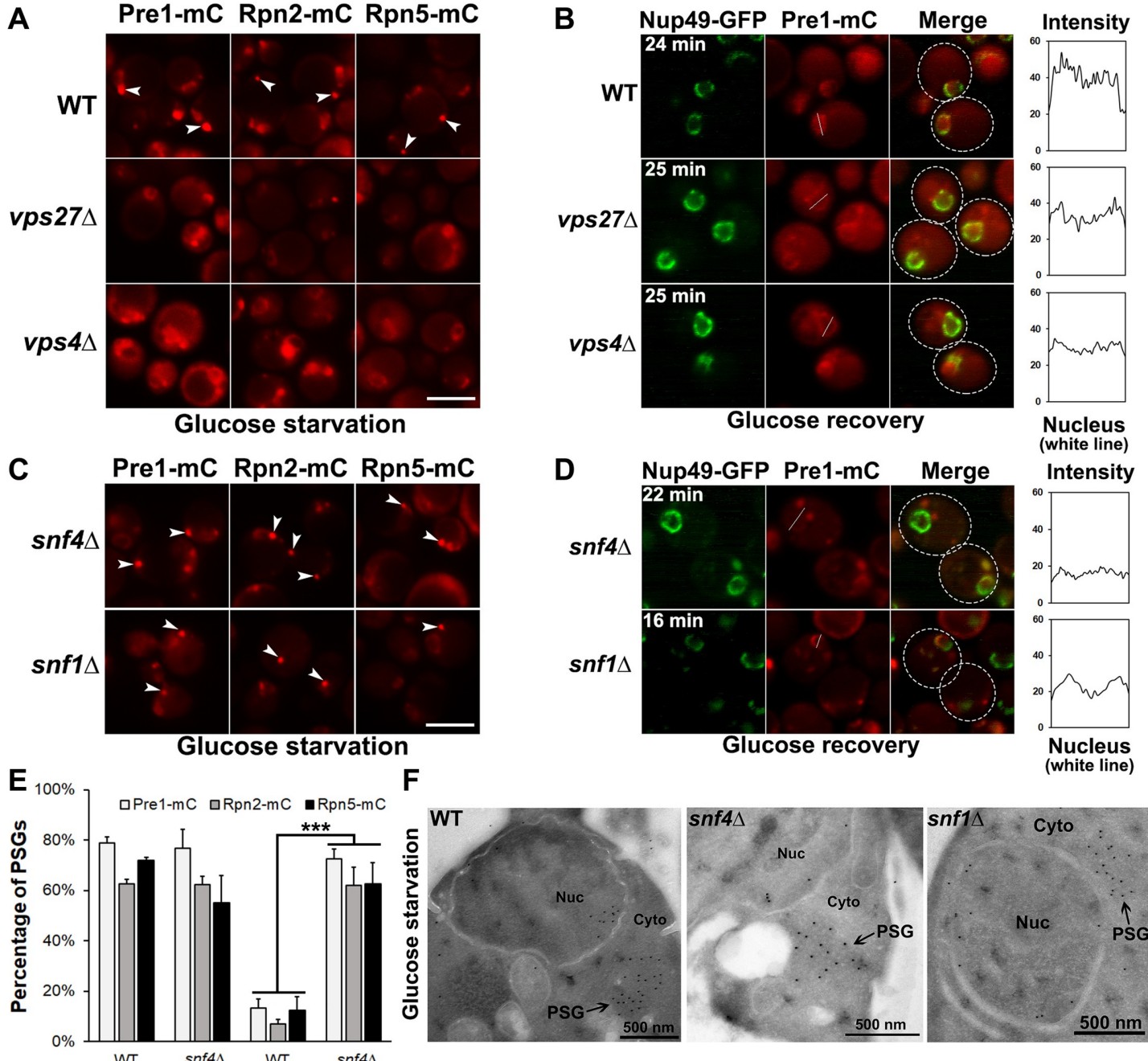

**Fig 1. AMPK and ESCRT proteins are required for reversible PSG assembly under glucose starvation.** (A) Epifluorescence microscopy images of Pre1-mC (mCherry) (a CP subunit, β4), Rpn2-mC (a base subunit), and Rpn5-mC (a lid subunit) in WT and ESCRT-defective mutant cells (*vps27Δ* and *vps4Δ*) during low glucose starvation. PSG assembly is largely disrupted in the mutants. (B) Confocal microscopy images of Pre1-mC in the ESCRT mutant cells after glucose recovery at the indicated time points. Defective nuclear reimport of proteasomes was also observed in the ESCRT mutants. (C) Epifluorescence images of Pre1-mC, Rpn2-mC, and Rpn5-mC in AMPK-defective cells (*snf4Δ* and *snf1Δ*) under low glucose. PSGs observed in AMPK mutants were comparable to those in WT cells. (D) Confocal images of Pre1-mC in the AMPK mutant cells after glucose recovery at the indicated time points. The PSGs in AMPK mutant cells failed to dissipate upon glucose refeeding. (E) Percentage of cells with PSGs visualized with Pre1-mC, Rpn2-mC, and Rpn5-mC in WT and AMPK mutant (*snf4Δ*) cells under low glucose and after 15 min glucose recovery. Cells counted in WT: Pre1-mC (263 in low glucose, 350 after glucose recovery), Rpn2-mC (389, 392), Rpn5-mC (349, 356) and *snf4Δ*: Pre1-mC (243, 317), Rpn2-mC (312, 247), Rpn5-mC (321, 227). Results plotted as mean±sd. ***, *P*<0.001 (ANOVA single factor analysis comparing *snf4Δ* to WT). (F) Cryo-immunogold electron micrographs of PSGs in the WT and AMPK mutant cells during glucose starvation. Cells were immunolabeled with anti-CP primary antibody and 10-nm Protein A-gold-bead-conjugated secondary antibody. Nuc: nucleus; Cyto: cytoplasm. Cells were harvested from cultures grown in SC medium containing 0.025% glucose for ~4 days at 30°C in panels (A), (C), and (F). White arrowheads point to PSGs in panels (A) and (C). Scale bars, 5 μm in panels (A) and (C). In panels (B) and (D), the times indicate when the

images were taken after glucose add-back; Nup49-GFP served as a nuclear envelope marker; white dotted circles indicate the approximate outlines of cells. The line charts quantify signal intensity of Pre1-mC in the indicated white line across the nucleus after glucose add-back.

found in the vacuole in WT, *snf4Δ*, and *snf1Δ* cells, but they were mostly adjacent to the vacuolar membrane in ESCRT mutants with the exception that Pre10-GFP was found in the vacuole lumen in *vps4Δ* cells (Fig 2B and 2C, S2A Fig). These localization data were consistent with the GFP immunoblot analysis (Fig 2A), indicating the ESCRT machinery is involved in proteasome trafficking to the vacuole during nitrogen starvation. Considering the role of the ESCRT machinery in sorting cargos to the lumen of MVBs, we propose that the ESCRT pathway is involved in sorting a fraction of proteasomes to the vacuole for degradation under these conditions.

## AMPK and ESCRT linked to proteasome subunit cleavage during glucose starvation

Autophagy is induced in cells when they are subjected to carbon-free starvation following pre-growth in various carbon sources with the notable exception of glucose [42]. Interestingly, autophagy is induced in cells pre-grown in glucose if they are transferred to low glucose (such as 0.05% glucose) medium rather than glucose-free medium [42]. The glucose starvation medium we used for PSG induction contained 0.025% glucose, so our protocol also likely induced autophagy.

Therefore, we assessed proteasome autophagy and whether it was affected in AMPK and ESCRT mutants under our conditions. We set up two glucose starvation regimens, culturing cells in either 0.025% glucose ("0.025% C" in figures) or glucose-free medium ("-C"). We found that in WT cells, proteasome subunit cleavage, specifically of the Pre10-GFP, Rpn5-GFP, and Rpn2-GFP subunits, was more prominent under low glucose compared to glucose-free conditions (compare WT lanes in top and bottom panels in Fig 3A). Protein fragments larger than free GFP were seen during carbon starvation, unlike what is observed during nitrogen starvation. Why these larger fragments accumulate is unknown but serves as a useful signature for proteasome degradation under carbon limitation.

Cleavage into these larger GFP-tagged fragments was inhibited in almost all the ESCRT mutants. Interestingly, only a minor defect was observed in *vps37Δ* cells, which affects ESCRT-I, whereas a much stronger defect occurred in *vps28Δ*, which alters another ESCRT-I component. This suggests ESCRT-I activity in proteasome degradation might be distinct from its action in the MVB pathway. Proteasome subunit cleavages were nearly completely abolished in the AMPK mutants (Fig 3A). These observations suggest AMPK plays a critical role in the regulation of proteasome degradation under low glucose conditions and implicate the ESCRT machinery as well.

Considering that proteasomes accumulated adjacent to the vacuolar membrane in ESCRT mutants under nitrogen starvation (Fig 2B and 2C, S2A Fig), we tested if they were at similar cellular sites in these mutants during glucose starvation. These could represent the aberrant stacked membranes called class E compartments in ESCRT mutants, such as those lacking *VPS4* [43]. We examined the localization of Pre1-mC, Rpn5-mC, and Rpn2-mC in *vps4Δ* cells expressing a vacuolar membrane protein marker, Vph1-GFP, in low glucose conditions. In addition to the diffusely localized proteasomes and proteasome granules in the cytoplasm, proteasomes colocalized with Vph1-GFP at class E-like compartments in ~42–51% of *vps4Δ* cells based on confocal microscopy (Fig 3B and 3C). Similar juxta-vacuolar localization of proteasomes was also found in other ESCRT mutants, although lower perivacuolar proteasome

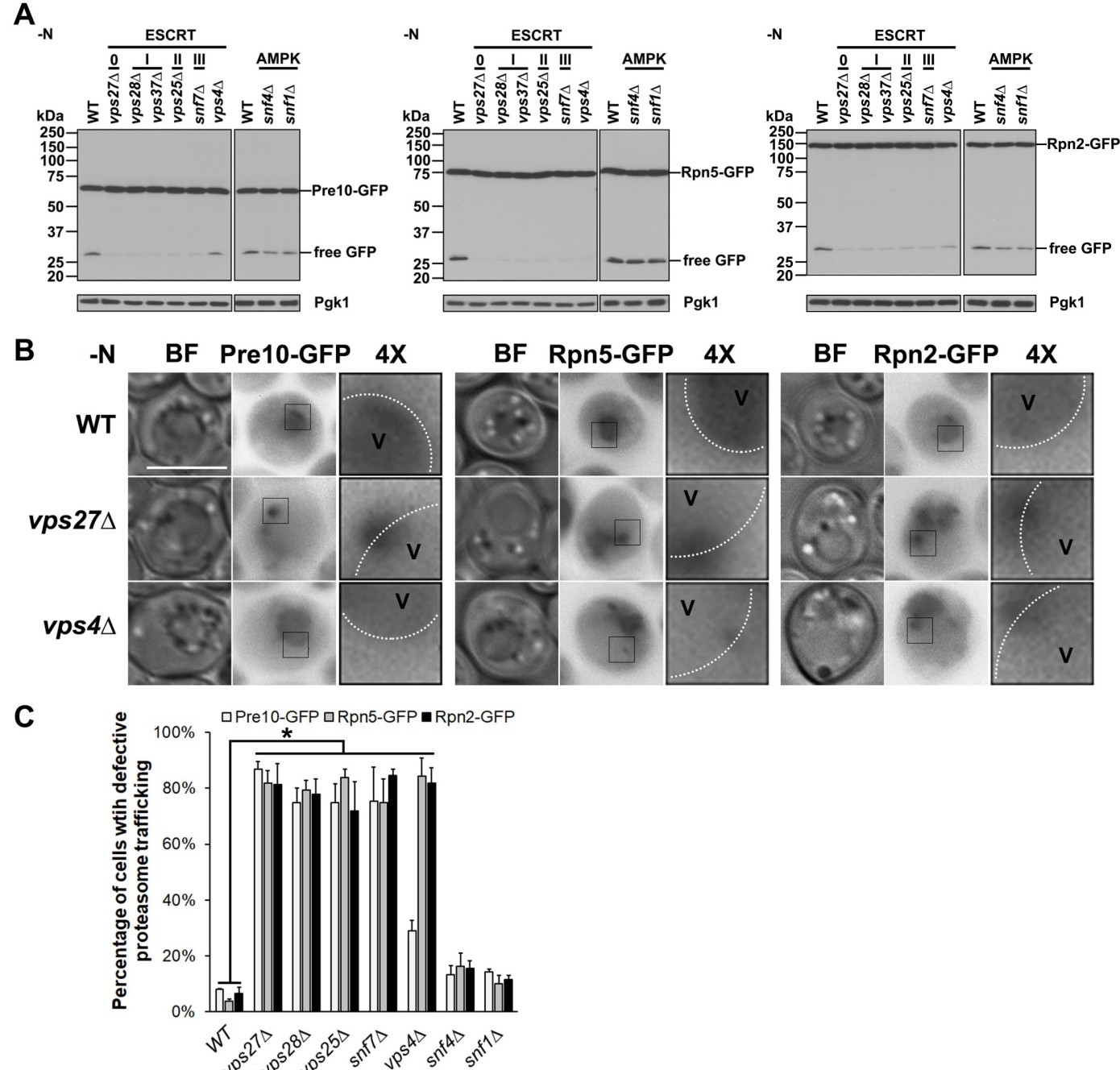

**Fig 2. Proteasome macroautophagy requires the ESCRT machinery but not AMPK under nitrogen starvation.** (A) Immunoblot analyses of Pre10-GFP (a CP subunit, α7), Rpn5-GFP, and Rpn2-GFP in WT, ESCRT mutant (*vps27Δ*, *vps28Δ*, *vps37Δ*, *vps25Δ*, *snf7Δ*, *vps4Δ*), and AMPK mutant (*snf4Δ*, *snf1Δ*) cells cultured in nitrogen starvation medium (-N) for ~1 day at 30°C. JL-8 monoclonal anti-GFP antibody was used to detect full-length proteasome subunit fusions and free GFP. Pgk1 served as a loading control. (B) Epifluorescence images of Pre10-GFP, Rpn5-GFP, and Rpn2-GFP in WT and ESCRT mutant (*vps27Δ* and *vps4Δ*) cells from panel (A). GFP-tagged proteasome signals were observed in the vacuole lumen in WT cells but adjacent to the vacuolar membrane (marked with dotted white line) in *vps27Δ* and *vps4Δ* cells under nitrogen starvation, with the exception that Pre10-GFP was observed in the vacuole lumen in *vps4Δ* cells. BF: bright field. V: vacuole. 4×: 4× enlargement of the squared regions in cells. Scale bar, 5 μm. Representative blots and images are shown. (C) Quantification of cells with defective proteasome trafficking to the vacuole in WT and mutant cells. The number of cells analyzed (for Pre10-GFP, Rpn5-GFP, Rpn2-GFP) were as follows: WT (226, 216, 338), *vps27Δ* (228, 256, 275), *vps28Δ* (327, 276, 232), *vps25Δ* (239, 236, 351), *snf7Δ* (209, 230, 226), *vps4Δ* (306, 288, 218), *snf4Δ* (259, 218, 318), and *snf1Δ* (255, 256, 356). Results plotted as mean±sd. *, P<0.05 (ANOVA single factor analysis comparing ESCRT mutants to WT).

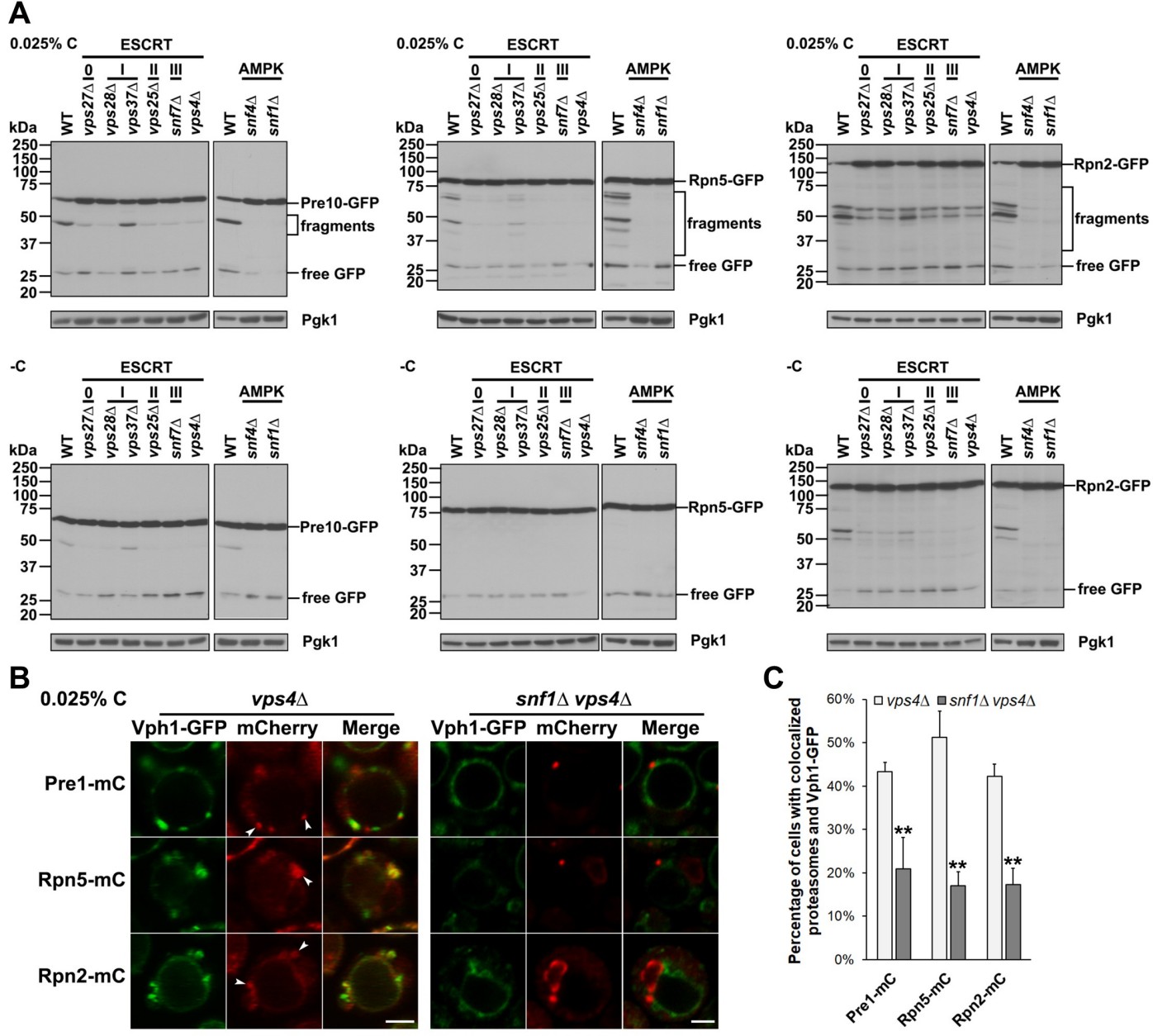

**Fig 3. AMPK and ESCRTs are required for proteasome fragmentation under glucose starvation.** (A) Anti-GFP immunoblot analyses of free GFP release and proteasome fragmentation of Pre10-GFP, Rpn5-GFP, and Rpn2-GFP in WT and ESCRT mutants (*vps27Δ*, *vps28Δ*, *vps37Δ*, *vps25Δ*, *snf7Δ*, *vps4Δ*) and AMPK mutants (*snf4Δ*, *snf1Δ*) under glucose starvation. Cells were harvested from cultures in SC medium containing 0.025% glucose (0.025% C) or lacking glucose (-C) for ~4 days at 30˚C. Low glucose conditions induced more proteasome fragmentation than did glucose-free starvation. (B) Confocal images of Pre1-mC, Rpn5-mC, and Rpn2-mC in an ESCRT-defective mutant (*vps4Δ*) and an AMPK-ESCRT double mutant (*snf1Δ vps4Δ*) expressing the vacuolar membrane marker Vph1-GFP. Cells were harvested from cultures in SC medium containing 0.025% glucose for ~1 day at 30˚C. White arrowheads point to proteasomes that were blocked at apparent class E compartments. Scale bars, 2 μm. (C) Percentage of living cells with colocalized proteasome subunits and Vph1-GFP in *vps4Δ* and *snf1Δ vps4Δ* cells from panel (B). Cells counted in *vps4Δ*: Pre1-mC (226), Rpn5-mC (252), Rpn2-mC (253) and *snf1Δ vps4Δ*: Pre1-mC (270), Rpn5-mC (280), Rpn2-mC (264). Results plotted as mean±sd. **, $P<0.01$ (ANOVA single factor analysis comparing *snf1Δ vps4Δ* to *vps4Δ*).

accumulation and milder effects on PSG formation and dissipation were observed in *vps37Δ* cells (S2B and S2C Fig). Interestingly, the colocalization between proteasomes and Vph1-GFP dropped to ~17–21% of cells in a *snf1Δ vps4Δ* double mutant (Fig 3B and 3C). These data

suggest the ESCRT machinery promotes AMPK-regulated proteasome sorting from the cytoplasm into the vacuole during glucose starvation.

## AMPK regulates IPOD association of CP-containing PSGs in glucose-starved cells

The perivacuolar IPOD compartment serves as a cellular protein quality control site by sequestering misfolded cytosolic proteins [44]. It also participates in proteasome quality control by distinguishing aberrant proteasomal subunits from normal ones during PSG formation [17]. To investigate whether the apparently irreversible proteasome aggregates that form in AMPK mutants are associated with the IPOD, we combined a gene encoding the IPOD marker Hsp42-GFP with *snf4Δ* or *snf1Δ* in strains also expressing Pre1-mC, Rpn2-mC, or Rpn5-mC. We measured the colocalization frequencies between mC-tagged proteasome subunits and Hsp42-GFP over seven days of glucose starvation.

We counted colocalization events as cells with at least one mC-tagged focus (PSG) that overlapped with Hsp42-GFP (IPOD) staining (Fig 4A, Pre1-mC). The mC-tagged proteasome subunits that marked PSGs colocalized with Hsp42-GFP in ~10% of WT cells during the 7-day time course under either low glucose or glucose-free conditions (Fig 4B and 4D, S3 Fig). This may represent a basal level of PSG maturation into or direct proteasome exchange with IPODs. Rpn5-mC or Rpn2-mC colocalized with Hsp42-GFP to a similar degree in the AMPK mutants (S3 Fig), suggesting that RP-containing PSGs associate with or mature into IPODs independently of AMPK signaling. Surprisingly, the fraction of cells with PSGs marked with Pre1-mC (CP) increasingly colocalized with IPODs in the AMPK mutants, reaching nearly 60% after seven days in low glucose (Fig 4B). Under glucose-free starvation, however, Pre1-mC-containing PSGs maintained a basal (~WT) level of association with Hsp42-GFP (Fig 4D). These data suggest that AMPK specifically regulates CP-containing PSG association with the IPOD under low glucose conditions and that the kinase transduces small differences in glucose concentration into alternative proteasome trafficking and quality control pathways.

In light of the strongly increased colocalization between Pre1-mC and Hsp42-GFP in AMPK mutants after extended low glucose incubation (Fig 4B), we examined their localization upon glucose refeeding. Pre1-mC foci did not dissipate, as noted above, and colocalization between Pre1-mC and Hsp42-GFP dropped only partially: by ~46% and ~34% in the *snf4Δ* and *snf1Δ* cells, respectively, after 30 min glucose recovery (Fig 4C). By comparison, colocalization dropped by ~79% in WT cells due to PSG dissipation and proteasome nuclear reimport. This suggests that a fraction of the CP-containing foci in AMPK mutants, which still colocalized with IPODs after glucose recovery, had either become 'terminal PSGs' or matured into IPODs. The expected difference between these two structures would be the greater compositional complexity of IPODs.

## Proteasome subunit cleavage in glucose-depleted cells requires vacuolar proteases but not macroautophagy

Proteasomes are targeted to the vacuole through macroautophagy when cells are grown in the absence of nitrogen or carbon, leading to free GFP release [26, 28]. To assess whether macroautophagy is necessary for the more complex proteasome fragmentation observed under low glucose conditions, we examined yeast strains lacking macroautophagy-related genes. Loss of *ATG8* eliminates all types of macroautophagy [45], while *ATG11* and *ATG39* are needed for specific classes of selective macroautophagy [18]. *ATG17* is required only for non-selective macroautophagy [20], and *ATG15* encodes an intravacuolar lipase [46]. Under low glucose conditions, anti-GFP immunoblot analysis showed that cleavages of Pre10-GFP, Rpn5-GFP,

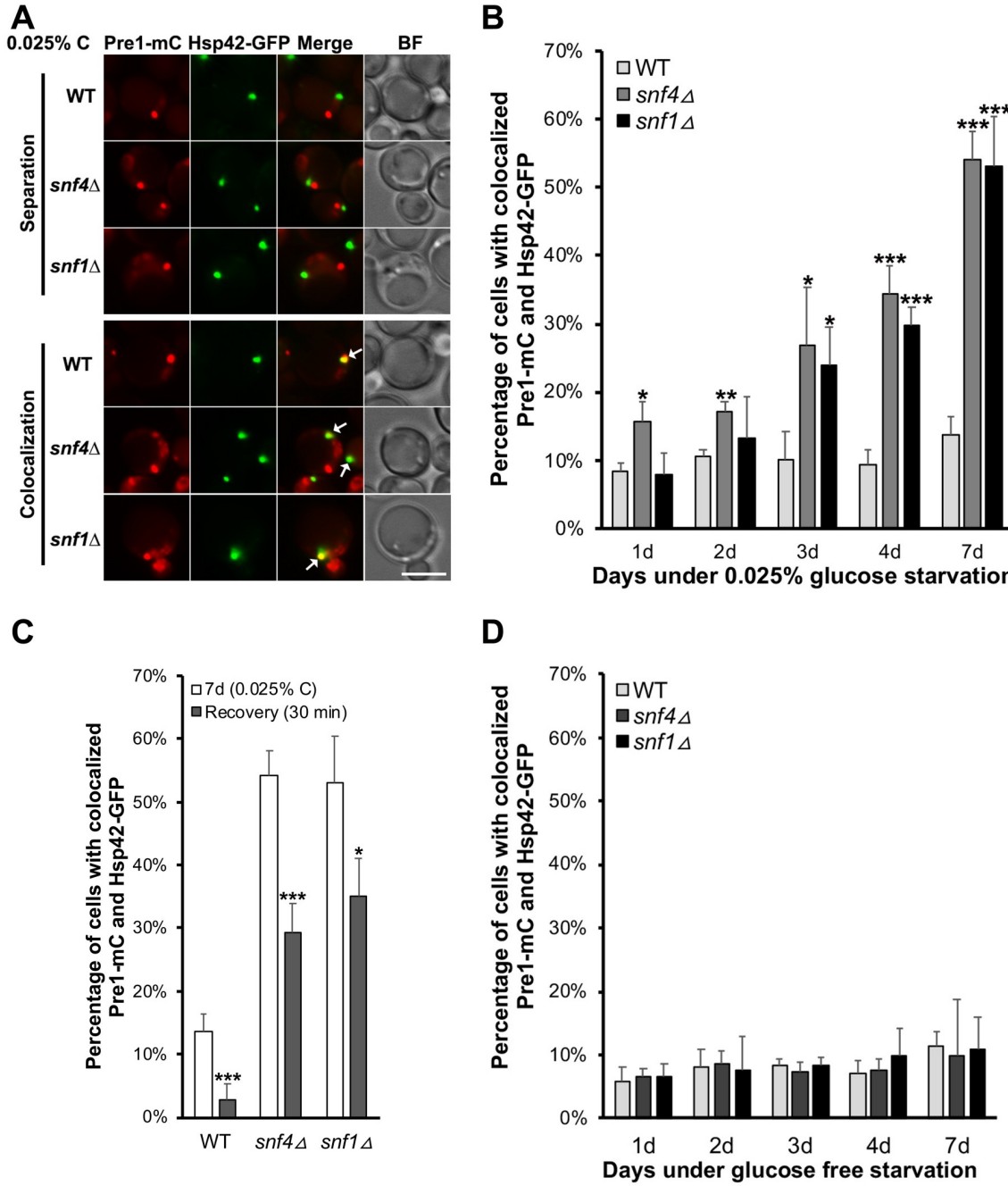

**Fig 4. CP-containing PSGs are enriched for IPOD marker Hsp42 in AMPK mutants in low glucose.** (A) Epifluorescence images of Pre1-mC and Hsp42-GFP in WT, *snf4Δ*, and *snf1Δ* cells in low glucose for ~4 days. Representative images of Pre1-mC where it separates from or colocalizes with Hsp42-GFP are shown. White arrows point to colocalized Pre1-mC and Hsp42-GFP. BF: bright field. Scale bar, 5 μm. (B) Quantification of colocalized Pre1-mC and Hsp42-GFP in live cells in 0.025% glucose for WT (438 cells counted [1d], 493 [2d], 338 [3d], 390 [4d], 546 [7d]), *snf4Δ* (n = 334 [1d], 845 [2d], 947 [3d], 652 [4d], 705 [7d]), and *snf1Δ* (n = 273 [1d], 551 [2d], 514 [3d], 531 [4d], 453 [7d]). (C) Quantification of colocalized Pre1-mC and Hsp42-GFP after 30 min recovery in 2% glucose in cells fixed after 7 d in 0.025% glucose from panel (B). Cells counted: WT (789), *snf4Δ* (692), *snf1Δ* (396). (D) Percentage of living cells with colocalized Pre1-mC and Hsp42-GFP. WT (433 cells counted [1d], 456 [2d], 693 [3d], 495 [4d], 355 [7d]), *snf4Δ* (519 [1d], 688 [2d], 473 [3d], 361 [4d], 220 [7d]), and *snf1Δ* (559 [1d], 484 [2d], 500 [3d], 412 [4d], 147 [7d]) cultures were grown in glucose-free medium. Results plotted as mean ±sd. *, $P<0.05$; **, $P<0.01$; ***, $P<0.001$ (ANOVA single factor analysis comparing *snf4Δ* or *snf1Δ* to WT).

and Rpn2-GFP within the proteasome subunit moieties of the fusions ("fragments" in Fig 5A) were not significantly different from WT cells in *atg8Δ*, *atg11Δ*, *atg17Δ*, *atg39Δ*, or *atg15Δ* mutants, but release of free GFP in these fusions was completely blocked in *atg8Δ* and *atg17Δ*

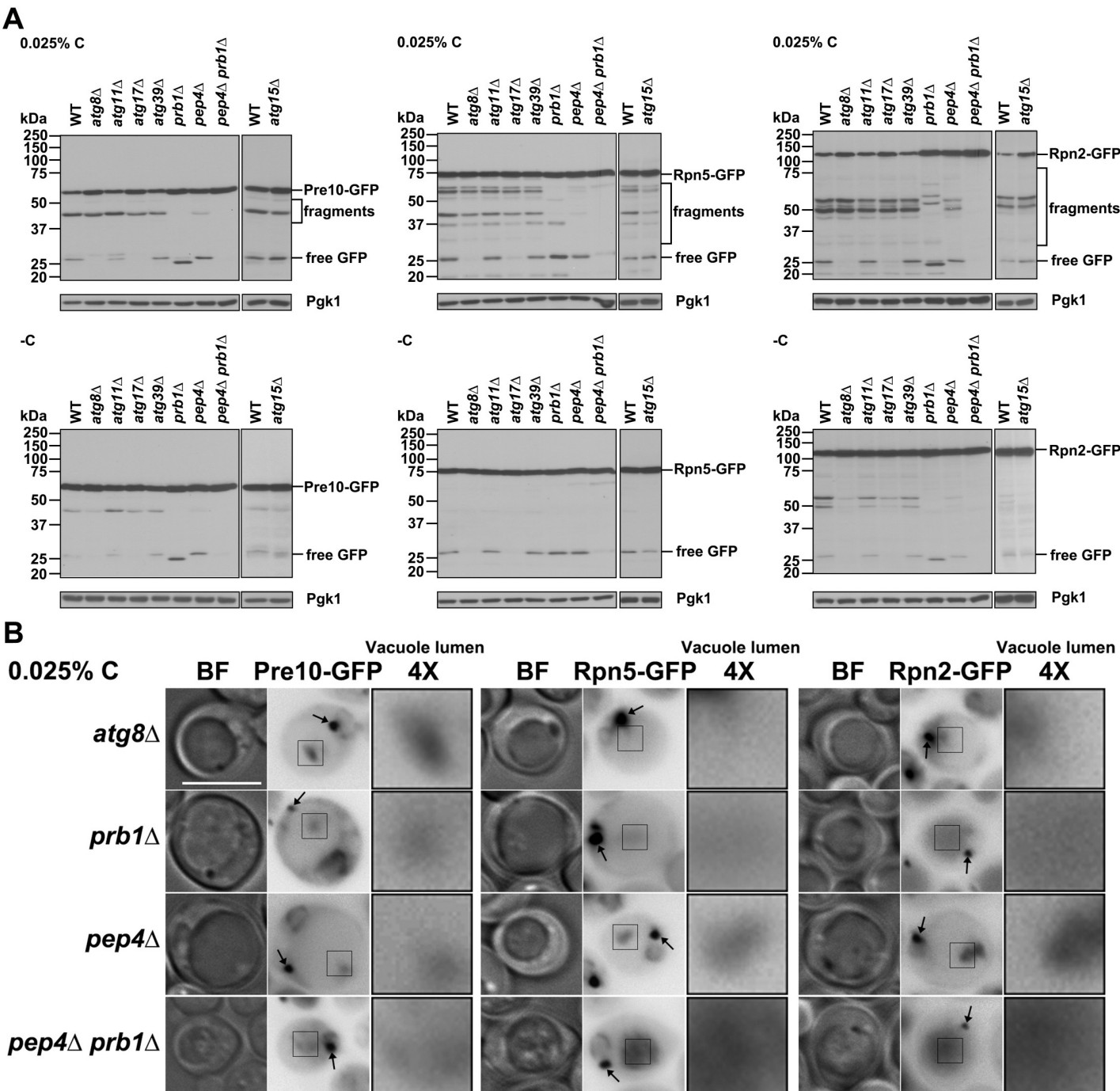

**Fig 5. Proteasome fragmentation in glucose-starved cells requires vacuolar proteases but not macroautophagy factors.** (A) Anti-GFP immunoblot analyses of WT, macroautophagy mutant (*atg8Δ*, *atg11Δ*, *atg17Δ*, *atg39Δ*, *atg15Δ*), and vacuolar protease-deficient mutant (*prb1Δ*, *pep4Δ*, *pep4Δ prb1Δ*) cells under glucose limitation. Cells were harvested from cultures in SC medium containing low glucose (0.025% C) or no glucose (-C) for ~4 days at 30°C. (B) Epifluorescence images of a core macroautophagy mutant (*atg8Δ*) and vacuolar protease-defective mutants from panel (A). Black arrows mark PSGs. BF: bright field. 4×: 4× enlargement of the square regions in the vacuole lumen. Scale bar, 5 μm.

cells (Fig 5A). PSGs continued to be observed in all of the tested *atg* mutants (Fig 5B and S4A Fig); upon glucose refeeding, the PSGs dissipated normally, and nuclear reimport of proteasomes was efficient (S4B Fig). By contrast, proteasome macroautophagy was blocked in nitrogen-starved *atg15Δ* cells (S4C Fig), in agreement with a previous study [28], although proteasome trafficking to the vacuole continued in the mutant, inasmuch as GFP-tagged full-length proteasome subunits accumulated in the vacuole lumen (S4D Fig). Collectively, these data suggest that complex proteasome fragmentation (in contrast to free GFP release), PSG assembly, PSG dissipation, and nuclear reimport of proteasomes are independent of macroautophagic components in cells cultured in low glucose.

To determine whether the complex proteasome subunit cleavages depended on vacuolar proteases, we introduced Pre10-GFP, Rpn5-GFP, and Rpn2-GFP into yeast strains lacking *PEP4*, encoding vacuolar protease A; *PRB1*, encoding protease B; or both genes, which blocks virtually all vacuolar proteolysis [47]. GFP immunoblot analysis showed that fragmentation of Pre10-GFP, Rpn5-GFP, and Rpn2-GFP was partially inhibited in the *prb1Δ* and *pep4Δ* single mutants and completely blocked in the *pep4Δ prb1Δ* double mutant during glucose deprivation (Fig 5A). PSGs formed normally in these mutants, along with retention of GFP-tagged proteasomes in the vacuole lumen (Fig 5B); PSG dissipation and nuclear reimport of proteasomes upon glucose refeeding was also similar to WT cells (S4B Fig). These results indicate that the proteasome subunit fragmentation requires vacuolar proteases and that these macroautophagy-independent cleavages could serve as a signature for an alternative mode of vacuolar import.

## ESCRT-dependent microautophagy is required for proteasome fragmentation

Another recently described mechanism for protein import and degradation in the yeast vacuole is ESCRT-dependent microautophagy [22]. Vph1 resides in the vacuolar membrane and is subjected to vacuolar degradation by microautophagy when cells are grown beyond the diauxic shift stage; this is independent of macroautophagy factors [22]. We first examined Vph1-GFP degradation under our three starvation conditions: nitrogen-free, glucose-free, and low glucose. Under nitrogen starvation, immunoblot analysis showed that free GFP release from Vph1-GFP was normal in *snf1Δ* and *snf4Δ* cells, while a minor reduction in degradation was seen in *atg15Δ* and *atg8Δ* cells; degradation was completely blocked in *vps4Δ* cells (Fig 6A). This suggests that under nitrogen starvation, vacuolar degradation of Vph1-GFP requires the ESCRT machinery but not AMPK, which recalled our data on AMPK and ESCRT dependence in proteasome macroautophagy under nitrogen starvation (Fig 2A and S4C Fig).

This contrasts with immunoblot analyses showing that free GFP release from Vph1-GFP was strongly inhibited in *snf1Δ* and *snf4Δ* cells as well as *vps4Δ* cells under both low glucose and glucose-free conditions. Interestingly, GFP release was also partially inhibited in *atg15Δ* and *atg8Δ* cells under glucose-free conditions, but was normal in these mutants in low glucose (Fig 6B). Therefore, vacuolar degradation of Vph1-GFP requires AMPK and the ESCRT pathway but is independent of macroautophagy under low glucose conditions. These results closely parallel the data on proteasome fragmentation under glucose starvation (Figs 3A and 5A). The implication is that like Vph1 cleavage, proteasome fragmentation in cells grown in low glucose conditions results from ESCRT-dependent microautophagy.

To observe proteasome microautophagy cytologically, we fused the GFP sequence to different proteasome subunit genes in a triple mutant lacking *ATG8*, *PEP4*, and *PRB1*; the latter mutations block all macroautophagy and most or all vacuolar hydrolysis reactions. Confocal microscopy of cells in low glucose suggested that PSGs marked by Pre10-GFP, Rpn5-GFP, or

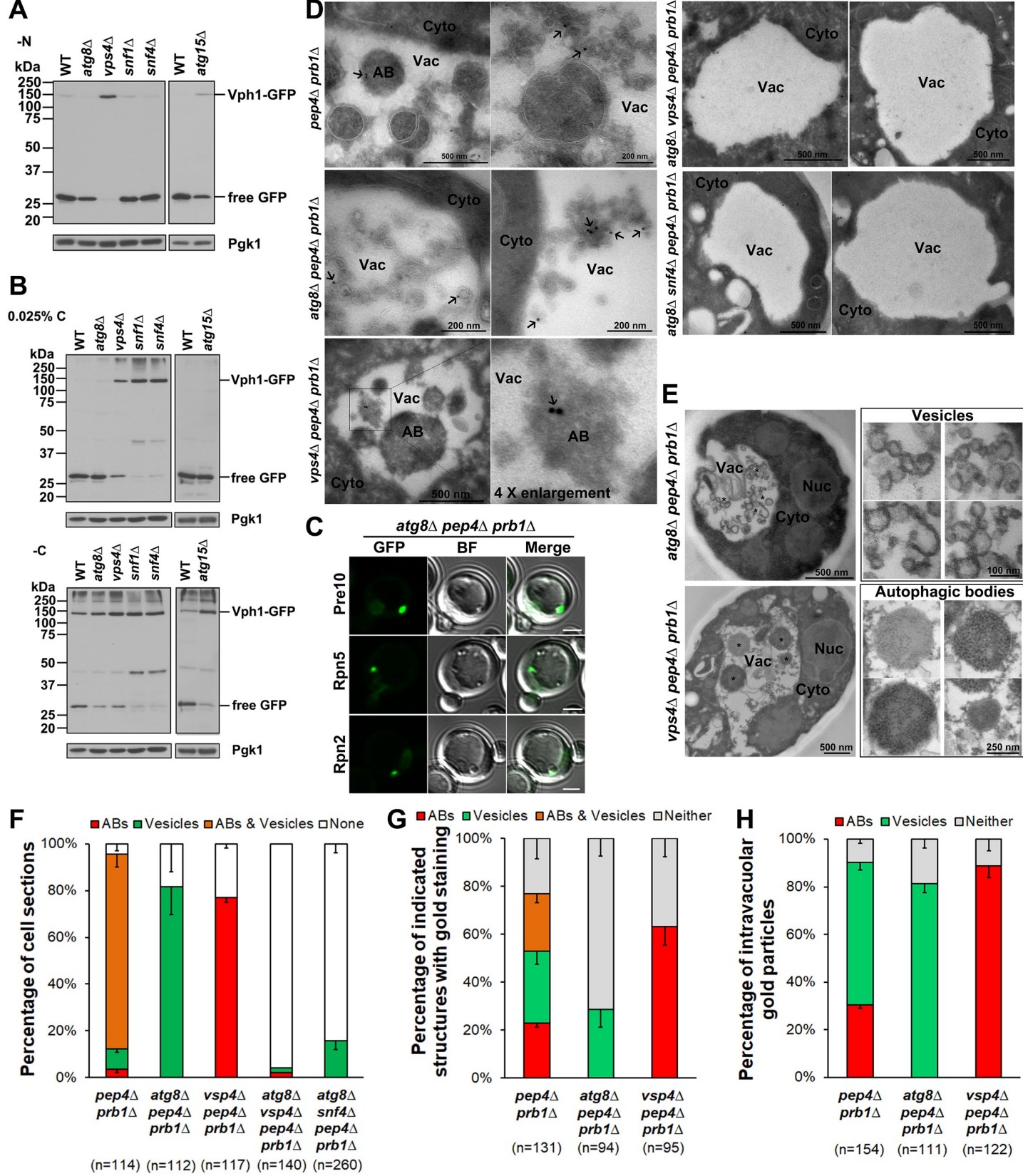

**Fig 6. Evidence for proteasome microautophagy in low glucose conditions.** (A) Anti-GFP immunoblot analysis of free GFP release from the vacuolar membrane protein Vph1-GFP in WT, macroautophagy mutant (*atg8Δ* and *atg15Δ*), ESCRT mutant (*vps4Δ*), and AMPK mutant (*snf4Δ* and *snf1Δ*) cells under nitrogen starvation for ~1 day at 30°C. (B) Immunoblot analyses of free GFP release from Vph1-GFP in the same strains as in (A) but during glucose starvation. Cells were harvested from cultures after ~4 days at 30°C. (C) Confocal microscopy images of Pre10-GFP, Rpn5-GFP, and Rpn2-GFP in *atg8Δ pep4Δ prb1Δ* cells in low glucose. Related videos are provided in the supporting information. BF: bright field. Scale bars, 2 μm. (D) Cryo-immunogold electron micrographs of proteasomes in the indicated cells grown in low glucose. Cells were immunolabeled with anti-CP primary antibody and Protein A-gold-conjugated secondary antibody (black arrows). (E) Electron micrographs of microautophagic vesicles and autophagic bodies formed in *atg8Δ pep4Δ prb1Δ* and *vps4Δ pep4Δ prb1Δ* cells, respectively, during low glucose starvation. "*" indicates highlighted phenotypes in left panels that enlarged as shown in right panels. Nuc: nucleus; Cyto: cytoplasm; Vac: Vacuole; AB: autophagic body. Cells were harvested from cultures in SC medium containing 0.025% glucose for ~1 day in panel (C) and ~4 days in panels (D) and (E) at 30°C. (F) Percentage of cell sections with the indicated intravacuolar structures (ABs, vesicles, ABs & vesicles, none) in the strains used for panel (D). (G) Percentage of the indicated structures with gold particle staining or neither AB nor vesicle (Neither) staining in the cells used for panel (D). (H) Percentage of intravacuolar gold particles staining either ABs or vesicles or neither in the cells used for panel (D). "n" represents cell sections counted in panels (F) and (G), and intravacuolar gold particles counted in panel (H).

Rpn2-GFP were tightly associated with vacuolar membrane invaginations (Fig 6C and S1–S3 Videos). Strikingly, by anti-CP immunogold electron microscopy, large numbers of membrane vesicle-like structures accumulated in the vacuole in ~82% of *atg8Δ pep4Δ prb1Δ* cell sections (Fig 6D and 6F); ~29% of vesicle-containing cell sections were marked with gold beads (Fig 6G). Roughly 81% of the intravacuolar gold beads marking proteasomes were on the edges of the membrane vesicle-like structures in *atg8Δ pep4Δ prb1Δ* cells (Fig 6D and 6H). The vesicles or tubules fill a large fraction of the vacuole lumen and were often delimited by apparent double membranes, possibly due to folding of invaginated membranes (Fig 6E). By contrast, in glucose-limited *vps4Δ pep4Δ prb1Δ* cells, which lack the full ESCRT pathway and therefore microautophagy, CP-labeled autophagic bodies (ABs) were observed in the vacuole lumen instead; ~77% of the cell sections included ABs (Fig 6E and 6F), and in these sections, ~63% were labeled with anti-CP gold beads, of which ~89% marked ABs (Fig 6D, 6G and 6H). This is consistent with the known role of macroautophagy in proteasome degradation when cells are carbon-starved [31] and our GFP release assays (Figs 3A and 5A). Moreover, the microautophagic vesicles and ABs were induced simultaneously as both structures were observed in ~83% of *pep4Δ prb1Δ* cell sections, while neither was visible in ~96% of *atg8Δ vps4Δ pep4Δ prb1Δ* and ~84% of *atg8Δ snf4Δ pep4Δ prb1Δ* cell sections (Fig 6D and 6F). These findings support the hypothesis that AMPK specifically regulates ESCRT-dependent microautophagy of proteasomes in low glucose conditions.

## Aberrant proteasomes are more prone to microautophagy during glucose starvation

Proteasome microautophagy might represent a type of protein triage or quality control under nutrient stress conditions. We hypothesized that aberrant proteasomes would sort into microautophagy structures rather than reversible PSGs in cells subjected to glucose limitation. To create cells bearing large numbers of misassembled or abnormal proteasomes, we used yeast strains lacking either of two nonessential proteasome subunits, Sem1/Rpn15 or Pre9/α3, which affect assembly of the RP and CP, respectively [48, 49].

Detection of free GFP and novel cleavage fragments from Pre10-GFP, Rpn5-GFP, and Rpn2-GFP was enhanced in *sem1Δ* and *pre9Δ* cells under both low glucose and glucose-free conditions (Fig 7A). At the same time, formation of PSGs from GFP-tagged CPs and RPs was abolished in the *sem1Δ* cells, as was PSG assembly of Pre10-GFP-tagged CPs in *pre9Δ* cells (Fig 7B). PSG assembly of Rpn2-GFP and Rpn5-GFP was significantly inhibited but not fully blocked in the *pre9Δ* mutant, consistent with formation of RP-only PSGs (Fig 7B and 7C). Rpn2-GFP-containing PSGs were observed in only ~36% of *pre9Δ* cells compared to ~75% of WT cells, and Rpn5-GFP-marked PSGs were observed in ~23% of *pre9Δ* cells versus ~79% of WT cells (Fig 7C). The PSGs of Rpn2-GFP and Rpn5-GFP still dissipated, and proteasomes

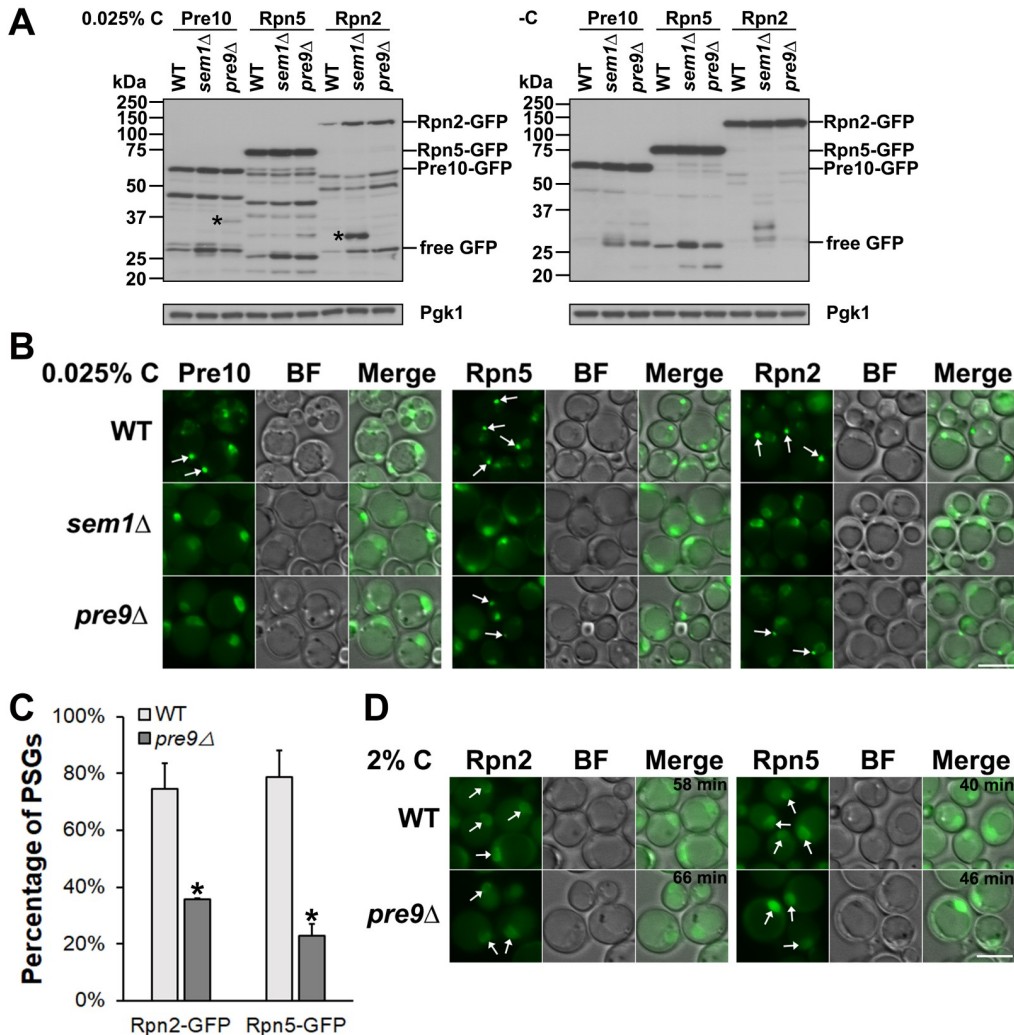

**Fig 7. Defective proteasomes are channeled to the vacuole rather than PSGs in glucose-starved cells.** (A) Anti-GFP immunoblot analyses of Pre10-GFP, Rpn5-GFP, and Rpn2-GFP in WT and proteasome mutant (*sem1Δ* and *pre9Δ*) cells during glucose starvation. Cells were cultured for ~4 days at 30°C in SC media with low or no glucose. "*" indicates novel fragments not seen in WT. (B) Epifluorescence images of Pre10-GFP, Rpn5-GFP, and Rpn2-GFP from the same cultures grown in low (0.025%) glucose from panel (A). CP-containing PSGs were no longer observed in *sem1Δ* or *pre9Δ* cells, and RP-containing PSGs were not formed in *sem1Δ* cells and were significantly reduced in *pre9Δ* cells. White arrows point to PSGs. (C) Percentage of cells with PSGs visualized with Rpn5-GFP and Rpn2-GFP in WT (793 cells counted [Rpn2-GFP], 890 [Rpn5-GFP]) and *pre9Δ* (938 cells counted [Rpn2-GFP], 819 [Rpn5-GFP]) cells. Results shown as mean±sd. *, $P<0.05$ (ANOVA single factor analysis). (D) Epifluorescence images of Rpn5-GFP and Rpn2-GFP in WT and *pre9Δ* cells after glucose add-back for the indicated times. Cultures were the same as in panel (B). White arrows point to nuclei with reimported proteasomes. BF: bright field. Scale bars, 5 μm in panels (B) and (D).

were reimported to the nucleus in the *pre9Δ* mutant after glucose recovery (Fig 7D). The novel fragments generated from RP subunit Rpn2-GFP and CP subunit Pre10-GFP specifically in the *sem1Δ* RP mutant and *pre9Δ* CP mutant, respectively, and the greater sensitivity of Pre10-GFP localization to PSGs in the *pre9Δ* CP mutant support the idea that CP and RP trafficking through PSGs is likely regulated through at least partly distinct mechanisms.

To test further the notion that aberrant proteasomes are channeled into the microautophagy pathway rather than assembled into PSGs during low glucose starvation, we tested if inhibition of CP catalytic activity by the drug MG132 affected proteasome fragmentation and

PSG assembly under these conditions. Immunoblot analysis showed that fragmentation of Pre6-GFP (a CP subunit, α4), Rpn5-GFP, and Rpn2-GFP was enhanced by MG132 compared to the DMSO control (S5A Fig). Strikingly, assembly of these tagged proteasomes into PSGs was completely blocked in MG132-treated cells (S5B Fig). These findings suggest that cells can recognize inactive or abnormal proteasomes and preferentially remove them by autophagy under low glucose conditions.

## Discussion

The results presented here demonstrate that subcellular proteasome trafficking and degradation are highly regulated during glucose limitation by the integrated action of AMPK (Snf1 kinase) and the ESCRT machinery (Fig 8). In particular, we discovered that AMPK promotes ESCRT-dependent microautophagy of proteasomes under low glucose conditions, and this degradation mechanism appears to be biased toward inactive or abnormal proteasomes. This selectivity allows functional proteasomes to accumulate in PSGs, which after glucose refeeding rapidly dissipate, with reimport of proteasomes into the nucleus. In the absence of AMPK, when ESCRT-dependent microautophagy is compromised, the IPOD serves as an alternative cellular site to sequester and remove proteasome granules that fail to disassemble normally, especially those containing only CP complexes.

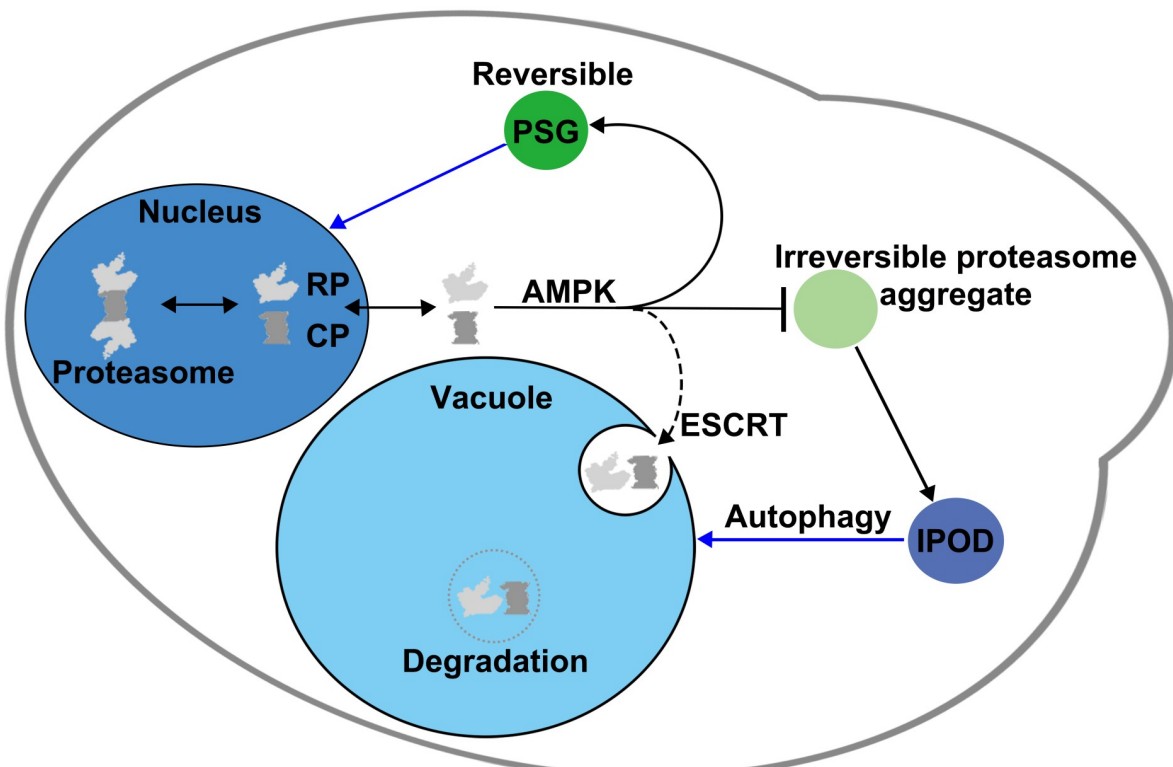

**Fig 8. Model for cellular control of proteasome trafficking and degradation during low glucose and glucose recovery.** In this working model, AMPK links glucose status with the trafficking and degradation of proteasomes. Normally, AMPK will either directly or indirectly promote formation of PSGs in low glucose conditions. A fraction of proteasomes will be inactive or defective after prolonged starvation, and ESCRT-dependent microautophagy preferentially targets this fraction for degradation in the vacuole. In AMPK mutants, proteasome microautophagy is blocked, and irreversible proteasome aggregates accumulate both aberrant and normal proteasomes. In this case, irreversible proteasome aggregates mature into or transfer proteasomes to the IPOD compartment to sequester potentially defective or inactive proteasomes, particularly irreversible CP-containing aggregates. These structures might be degraded by macroautophagy or, upon glucose refeeding and re-entry into the cell cycle, might utilize retention in mother cells as a means to clear aberrant proteasomes.

## Independent regulation of CP and RP during nutrient limitation

In proliferating yeast and most mammalian cells, proteasomes concentrate in the nucleus, but large-scale proteasome relocalization from the nucleus to other intracellular compartments is triggered by nutrient limitation. During nitrogen starvation, proteasomes are routed by macroautophagy to the vacuole for degradation, whereas they mostly relocate to the cytoplasm into large, membraneless granules (PSGs) during carbon starvation.

In nitrogen-starved cells, nuclear proteasomes have been reported to disassemble into CP and RP lid and base subcomplexes [28], and the CP and RP are regulated by different mechanisms during their trafficking from the nucleus to the vacuole. For example, CP macroautophagy depends on the deubiquitylating enzyme Ubp3, but RP macroautophagy does not [27]. Here we have shown that many components of the ESCRT pathway are required for proteasome macroautophagy, probably by facilitating the sealing of proteasome-containing autophagosomes; ESCRT factors have been implicated directly in this stage of macroautophagy [50]. It would be reasonable to suggest a similar function for ESCRTs during vacuolar membrane vesiculation during microautophagy. Surprisingly, the Vps4 AAA-ATPase, while required for RP macroautophagy, is not essential for CP macroautophagy, suggesting different roles for some ESCRT factors in proteasome trafficking and degradation. There are other cellular processes, such as cytokinesis, where ESCRT factors also function differently than in the MVB pathway [51]. Vps4 disassembles and recycles the ESCRT-III polymers responsible for membrane fission/fusion during autophagosome closure [52]. It is possible that Vps4 activity in CP macroautophagy can somehow be replaced by another ATPase [52, 53]. The molecular mechanisms of ATPase action in CP macroautophagy during nitrogen starvation will require further investigation.

Reversible cytoplasmic PSGs are assembled under glucose starvation in yeast and plants [13, 31]. Previous studies reported that proteasomes also disassemble into CP and RP during prolonged incubation in stationary phase (which is also a form of carbon starvation) or under carbon starvation [31, 54], but CP and RP eventually accrete into the same 1–2 large PSGs per cell [13, 17]. We have found that mutations in either the CP or RP inhibit PSG assembly (Fig 7B–7D). Alteration of the lid by deletion of *SEM1* completely blocked PSG formation from CP, base, and lid subcomplexes (Fig 7B). This is consistent with previous studies showing that the integrity of lid subunits (Rpn5 and Rpn11) is required for PSG assembly [12, 16, 17]. Interestingly, creation of an alternative form of the CP by deletion of *PRE9* completely blocked CP incorporation into PSGs while incompletely inhibiting PSG assembly from base and lid components (Fig 7B–7D). This supports the inference that proteasomes separate into CP and RP complexes during glucose starvation, and further suggests that PSGs assemble from RP complexes alone in cells expressing a defective CP, analogous to previous observations in *blm10Δ* cells [15].

## ESCRT-dependent microautophagy versus reversible PSG assembly

Our data suggest that AMPK and ESCRT-dependent microautophagy play an important role in sorting and eliminating defective proteasomes in cells starved for glucose. Complex proteasome fragmentation is more strongly induced under low glucose (0.025% glucose) than glucose-free conditions (Figs 3A, 5A and 7A) and requires AMPK; the ESCRT machinery also contributes to this fragmentation by vacuolar proteases (Figs 3A and 5A), but it is largely independent of macroautophagy factors (Fig 5A). ESCRT-dependent microautophagy during glucose starvation was only recently demonstrated [22]. Our data show in addition that AMPK plays a critical role in microautophagy of the vacuolar membrane protein Vph1, the primary model substrate used in the earlier study (Fig 6B).

In light of the similar requirements for proteasome and Vph1 degradation in response to glucose starvation, we surmised that proteasome subunit fragmentation could also be through a type of microautophagy under low glucose conditions. We found direct evidence of proteasome microautophagy by immunogold EM labeling of CPs (and time-lapse analysis of both CP and RP by confocal microscopy) in *atg8Δ pep4Δ prb1Δ* cells, which accumulate intravacuolar agglomerations of membrane tubules and vesicles that stain with anti-CP antibodies (Fig 6D). These cells, which cannot carry out macroautophagy and are incapable of degrading their vacuolar contents, collect enormous amounts of this intravacuolar material (Fig 6E). Proteasomes are mostly associated with the apparent double membranes of the tubules and vesicles in the vacuole (Fig 6D). Under conditions of extended glucose deprivation, which would be expected to occur in the wild, microautophagy is likely to be a highly active means of cellular remodeling and protein quality control, including of the proteasome.

The finding that aberrant proteasomes appear to be selectively cleared by microautophagy in low-carbon conditions and at the same time compromise reversible PSG assembly (Fig 7 and S5 Fig) raises the question of how abnormal or inactive proteasomes are distinguished and sorted into the degradation pathway. In fission yeast, the ESCRT machinery can mediate the selective autophagy of soluble ubiquitylated cargos via the MVB pathway [55]. This is independent of the core macroautophagy machinery [55]. Moreover, by examining the degradation of a vacuolar lysine transporter, Ypq1, a recent study reported that the ESCRT machinery functions directly on the vacuolar membrane for sorting ubiquitylated Ypq1 to the vacuole lumen [56]. We propose that aberrant or inactive proteasomes, which are known to get ubiquitylated [26], are sorted for microautophagy in a similar fashion.

Among the ESCRT proteins, Vps27, a component of the ESCRT-0 complex, recognizes ubiquitylated endosomal cargos and recruits ESCRT-I, II, and III for delivering cargos to the vacuole for degradation [57]. It can also work directly at the vacuolar membrane [56]. At the vacuolar membrane, Vps27 may recognize aberrant ubiquitylated proteasomes and initiate ESCRT assembly, leading to the selective elimination of these proteasomes by microautophagy. Proteasome ubiquitylation may occur at the vacuolar membrane; both our fluorescence microscopy data on PSG localization (Figs 4A, 5B and 7B and S2B Fig) and prior studies [17] have shown that the majority of PSGs localize adjacent to the vacuolar membrane during PSG assembly. Therefore, aberrant proteasomes might be accessible to E3 ubiquitin ligases on the vacuolar surface even if initially sorted into PSGs; ubiquitylated particles could then be sorted via the ESCRT machinery to the vacuole lumen. This model posits a direct proteasome quality-control mechanism at the PSG-vacuole interface, an idea we are currently testing.

## AMPK regulation of proteasome trafficking and degradation

Mammalian AMPK can promote autophagy by phosphorylation of the serine/threonine kinase Ulk1 [39, 40]. Ulk1 is a homolog of yeast Atg1 kinase, which is pivotal in macroautophagy initiation during nitrogen starvation [58]. But how does AMPK regulate proteasome microautophagy? Recently, microautophagy has been classified into three types according to the overt membrane dynamics (morphology and location) involved. These are microautophagy with lysosomal protrusion, with lysosomal invagination, or with endosomal invagination [59]. Proteasome microautophagy in our model (Fig 8), which is based on our cell biological data, is akin to vacuolar (lysosomal) invagination, although we cannot completely exclude endosomal invagination.

In microautophagy with vacuolar invagination, distinct vacuolar membrane domains, classified as liquid-ordered ($L_o$) and liquid-disordered ($L_d$), facilitate the invagination of vacuolar membranes in response to glucose depletion [60]. When cells enter stationary phase, for

example, lipid droplets (LDs) undergo microautophagy and likely enter the vacuole lumen through sterol-enriched $L_o$ domains [61]. AMPK is activated in response to low glucose levels and promotes the redistribution of Atg14 from ER exit sites onto $L_o$ domains to initiate LD microautophagy [62]. This might be relevant to PSG-vacuole association, allowing aberrant proteasomes to partition into lipid domains where the ESCRT machinery initiates proteasome microautophagy. In another scenario, AMPK may directly phosphorylate ESCRT proteins or proteasome subunits and thus enhance interaction of ESCRT factors and proteasomes.

### Regulation of proteasome trafficking by the IPOD during glucose starvation

Over a 7-day period of glucose limitation, we found a steady state of ~10% colocalized CP and RP foci with IPODs in WT cells (Fig 4 and S3 Fig). Consistent with our results, a previous study, while reporting increased colocalization between PSGs and IPODs at an early stage of PSG assembly during glucose starvation, observed that the majority of proteasomes eventually separated from the IPOD, resulting in ~10% of cells showing colocalization [17]. We also saw that in the absence of AMPK, the IPOD selectively sequesters excess CP granules that form during low glucose starvation but fail to dissipate upon glucose restoration in these mutants (Figs 4B, 4C and 8). The IPOD may thus serve as an alternative means of regulating proteasome homeostasis by collecting aggregated proteasomes that escaped from AMPK and ESCRT-dependent proteasome quality controls during PSG formation.

In summary, our findings extend the developing view that proteasomes, which are key components of the cellular protein homeostasis network, are themselves subject to diverse surveillance mechanisms when cells experience nutritional and other stresses. In glucose-limited cultures, yeast cells relocate most proteasomes to 1–2 large cytoplasmic foci, and these PSGs are subject to various types of dynamic control. Our data reveal a novel AMPK- and ESCRT-regulated pathway of proteasome microautophagy under these conditions that preferentially removes abnormal or inactive proteasomes from the cellular pool. How proteasomes are exchanged between PSGs and free proteasome pools both in the cytoplasm and nucleus, and how microautophagy, macroautophagy, and IPOD sequestration control these proteasome populations under different conditions will be important questions to pursue.

## Materials and methods

### Construction of an Rpn5-GFP integrated yeast deletion library

The construction of the *RPN5-GFP*-integrated yeast deletion library has been described previously [14]. Briefly, a benchtop colony arrayer (RoToR; Singer Instruments) was used to manipulate the yeast strains in high-density format (384-well plates). *RPN5-GFP* was introduced into the yeast deletion library by an SGA approach [36]. Specifically, a *MATα* haploid strain harboring *RPN5-GFP*::*URA3* and the required markers for SGA (*can1Δ*::*STE2pr-his5* and *lyp1Δ*) was mated with the deletion library. The resulting diploid cells were sporulated, and then haploid cells with the desired combination of *RPN5-GFP*::*URA3* and library gene deletion were selected.

### High-throughput fluorescence microscopy

Microscopic screening was performed using an automated microscopy setup as previously described [37]. Yeast cells were transferred from agar plates into liquid 96-well polystyrene growth plates using the RoToR arrayer. To induce PSGs, liquid cultures diluted to ~0.25 $OD_{600}$ (optical density unit at 600 nm) were grown in synthetic complete medium with no

glucose for 72 h in a shaking incubator (LiCONiC Instruments) at 30˚C. The liquid handler (JANUS; PerkinElmer) was then used to transfer strains into glass-bottom 96-well microscope plates (MatriCal Bioscience) coated with Concanavalin A (Sigma-Aldrich) to allow cell adhesion. Wells were washed twice to remove floating cells. To uncover mutants that interfered with PSG dissipation and nuclear reimport of proteasomes upon addition of glucose, synthetic defined medium containing glucose was added in the second wash. Plates were then transferred into an automated inverted fluorescent microscopic scan^R system (Olympus) using a swap robotic arm (Hamilton Robotics). The scan^R system is designed to allow autofocus and imaging of plates in 96-well format. Images of cells were recorded ~10 min following glucose replenishment at 24˚C using a 60× air lens (NA0.9) and with a cooled charge-coupled device camera (ORCA-ER; Hamamatsu Photonics). Images were acquired in the GFP channel (excitation at 490/20 nm and emission at 535/50 nm) and then manually visualized to uncover strains in which the normal re-localization of Rpn5 into the nucleus was disturbed. All images were taken at a single focal plane.

## Yeast strains and cell growth

Yeast manipulations were performed according to standard protocols [63]. Yeast strains used in this study are listed in S2 Table. Yeast cells were grown overnight in synthetic complete (SC) medium [42] at 30˚C with vigorous agitation. Cells were then diluted in fresh SC medium and grown to mid-log phase. Mid-log cells were pelleted, washed once with sterile $H_2O$, followed by different treatments. For nitrogen starvation, cells were resuspended in synthetic minimal medium lacking nitrogen (0.17% [w/v] yeast nitrogen base without amino acids and ammonium sulfate, 2% glucose) and cultured for ~1 day at 30˚C. For glucose starvation, cells were resuspended in SC medium containing 0.025% glucose or lacking glucose, and cultured for ~4 days at 30˚C. For proteasome inhibitor MG132 treatment, cells harboring a *PDR5* deletion (which allows efficient intracellular accumulation of the drug) were grown in SC medium as above, and mid-log cells were resuspended in SC medium containing 0.025% glucose and DMSO or 50 μM MG132 (Santa Cruz Biotechnology, catalog # sc-201270, lot # A1118) dissolved in DMSO and cultured for ~1 day at 30˚C. For glucose recovery, starved cells were washed once with sterile $H_2O$ and resuspended in SC medium (2% glucose) and cultured at room temperature (RT) for microscopy.

## Fluorescence microscopy

For epifluorescence microscopy, yeast cells were visualized on an Axioskop microscope (Carl Zeiss) equipped with a plan-Apochromat 100×/1.40 oil DIC objective lens equipped with a CCD camera (AxioCam MRm; Carl Zeiss) and a HBO100W/2 light source. Images were taken using AxioVision software. Chemical fixation of cells in Figs 1E and 4C was performed as described previously with minor modifications [64]. Briefly, glucose-starved cells were recovered in SC medium for 15 min or 30 min at RT and fixed with 2% (v/v) formaldehyde and incubated for 5 min at RT. The fixed cells were pelleted and washed once with 0.1 M $KPO_4$ pH 6.5, and then resuspended in 0.1 M $KPO_4$ pH 7.5 for imaging with a fluorescence microscope as above.

For confocal microscopy and time-lapse videos, yeast cells were viewed on an LSM 880 Airyscan NLO/FCS confocal microscope with an Alpha Plan-Apochromat 100×/1.46 NA oil objective lens. Excitation was performed with an argon laser at 488 nm and DPSS laser at 561 nm, and emission was collected in the range of 493–556 nm or 579–624 nm for GFP and mCherry imaging, respectively. Images were acquired using ZEN software and processed using Adobe Photoshop CS6 software.

## Electron microscopy

For immunogold labeling EM: yeast cells grown in SC medium containing 0.025% glucose for ~4 days were collected and fixed with 4% paraformaldehyde (PFA) and 0.2% glutaraldehyde in PBS for 30 min followed by further fixation in 4% PFA for 1 h. The fixed cells were rinsed with PBS, and resuspended in 10% gelatin. The blocks were trimmed and placed in 2.3 M sucrose on a rotor overnight at 4°C, and then transferred to aluminum pins and frozen rapidly in liquid nitrogen. The frozen blocks were cut on a Leica Cryo-EM UC6 UltraCut, and 60 nm thick sections were collected using the Tokuyasu method [65] and placed on carbon/Formvar-coated grids and floated in a dish of PBS for immunolabeling. Grids were placed section side down on drops of 0.1 M ammonium chloride to quench untreated aldehyde groups, then blocked for nonspecific binding on 1% fish skin gelatin in PBS. Single labeled grids were incubated with a primary antibody rabbit anti-20S (Enzo Life Sciences, catalog # BML-PW9355) at a dilution of 1:200, and 10 nm Protein A-gold (Utrecht Medical Center) was used as a secondary antibody. All grids were rinsed in PBS, fixed with 1% glutaraldehyde for 5 min, rinsed again and transferred to a uranyl acetate/methylcellulose drop before being collected and dried.

For regular EM: yeast cells grown in SC medium containing 0.025% glucose for ~4 days were collected and fixed with 2.5% glutaraldehyde and 2% PFA in phosphate buffer pH 6.9 for 1 h, then rinsed in PBS followed by rinsing in 0.1 M sodium acetate buffer pH 6.1. The cells were post fixed in 2% osmium in 0.1 M sodium acetate for 30 min at RT, rinsed with distilled water and resuspended in aqueous 2% uranyl acetate for 60 min at RT in the dark. Cells were then rinsed in distilled water and dehydrated in 95% and 100% ethanol. The pelleted cells were resuspended in propylene oxide, then infiltrated using LX112 (Ladd) epon resin and baked overnight at 60°C. Hardened blocks were cut using a Leica UltraCut UC7 and 60 nm sections were collected and stained using 2% uranyl acetate and lead citrate.

Grids were all viewed under a transmission electron microscope (FEI Tecnai G2 Spirit BioTWIN) at 80 kV. Images were taken using a SIS Morada 11-megapixel CCD camera and iTEM (Olympus) software. Acquired images were processed using Adobe Photoshop CS6 software.

## Protein extraction and Western blotting

Total proteins were extracted by the alkaline lysis method [66], and Western blotting was performed as described previously with minor modifications [64]. Cells equivalent to one $OD_{600}$ were collected by centrifugation and washed once with sterile $H_2O$. Cells were resuspended in 400 μl 0.1 M NaOH and incubated for 5 min at RT. Cells were pelleted, resuspended in 100 μl SDS sample buffer (10% glycerol, 2% SDS, 0.1 M DTT, 62.5 mM Tris-HCl pH 6.8, 4% 2-mercaptoethanol, 0.008% bromophenol blue) and heated at 100°C for 5 min. Cell debris were pelleted by centrifugation.

Equal volumes of the supernatants were loaded onto 10% (v/v) SDS-PAGE gels, followed by the transfer of proteins to polyvinylidene difluoride (PVDF) membranes (EMD Millipore, catalog # IPVH00010, lot # R8EA4245). The membranes were incubated with the primary antibodies Living Colors A.v. monoclonal antibody (JL-8; anti-GFP) (TaKaRa, catalog # 632381, lot # A5033481) at 1:2,000 dilution or anti-Pgk1 monoclonal antibody (Invitrogen, catalog # 459250, lot # TG2598062B) at 1:10,000 dilution, followed by ECL anti-mouse IgG secondary antibody conjugated to horseradish peroxidase (GE Healthcare, catalog # NA931V, lot # 16889300) at 1:10,000 dilution. The membranes were incubated in ECL detection reagent [67], and the protein signals were detecting using autoradiography film (Thomas Scientific, catalog # E3018).

## Cell viability analysis

Cell viability analysis (S7 Fig) was performed with propidium iodide (PI) staining. Low glucose-starved yeast cells at the indicated days were pelleted and resuspended in 0.1 ml PI/RNase staining buffer (BD Pharmingen, catalog # 550825), and incubated for 20 min at RT in the dark before imaging.

## Statistical analysis

ANOVA single factor analysis was performed using Microsoft Excel software. The number of cells counted for each assay are shown in the figure legends. Each experiment was repeated at least three times and the percentages shown in the figures represent the average of all the experiments. Error bars represent standard deviations.

## Supporting information

**S1 Fig. ESCRT proteins are required for reversible PSG assembly under glucose starvation.** (A) Epifluorescence images of Pre1-mC, Rpn2-mC, and Rpn5-mC in ESCRT-defective mutant cells (ESCRT-II: *vps25Δ*, and ESCRT-III: *did2Δ*, *vps2Δ*, *vps24Δ*, *snf7Δ*) during low glucose starvation. Scale bar, 5 μm. (B) Confocal images of Pre1-mC in the ESCRT mutant cells after glucose recovery at the indicated time points. The times indicate when the images were taken after glucose add-back; Nup49-GFP served as a nuclear envelope marker; white dotted circles indicate the approximate outlines of cells. The line charts quantify signal intensity of Pre1-mC in the indicated white line across the nucleus after glucose recovery.
(TIF)

**S2 Fig. ESCRT machinery is required for vacuolar trafficking of proteasomes during nitrogen or glucose starvation.** (A) Epifluorescence images of Pre10-GFP, Rpn5-GFP, and Rpn2-GFP in nitrogen-starved ESCRT mutant (*vps28Δ*, *vps25Δ*, and *snf7Δ*) and AMPK mutant (*snf4Δ*, *snf1Δ*) cells from figure panel (2A). (B) Epifluorescence images of Pre10-GFP, Rpn5-GFP, and Rpn2-GFP in low glucose-starved WT and ESCRT mutant cells from figure panel (3A). The vacuolar membrane is marked with dotted white line. BF: bright field. V: vacuole. 4×: 4× enlargement of the squared regions in cells. Scale bars, 5 μm. (C) Quantification of cells with abnormal perivacuolar proteasomes in WT and mutant cells in low glucose used in panel (B). Cells counted (Pre10-GFP, Rpn5-GFP, Rpn2-GFP): WT (278, 273, 391), *vps27Δ* (274, 265, 434), *vps28Δ* (437, 386, 253), *vps37Δ* (359, 323, 426), *vps25Δ* (312, 246, 319), *snf7Δ* (233, 253, 325), and *vps4Δ* (301, 336, 437). Results plotted as mean±sd. *, $P<0.05$ (ANOVA single factor analysis comparing ESCRT mutants to WT).
(TIF)

**S3 Fig. RP-containing irreversible proteasome aggregates maintain a basal colocalization level with IPODs in glucose-starved AMPK mutants.** (A) Quantification of colocalized Rpn5-mC and Hsp42-GFP in WT (577 cells counted [1d], 542 [2d], 748 [3d], 792 [4d], 446 [7d]), *snf4Δ* (546 [1d], 790 [2d], 838 [3d], 368 [4d], 524 [7d]), and *snf1Δ* (316 [1d], 656 [2d], 530 [3d], 374 [4d], 266 [7d]) live cells in 0.025% glucose. (B) Percentage of living cells with colocalized Rpn5-mC and Hsp42-GFP. WT (292 cells counted [1d], 580 [2d], 457 [3d], 419 [4d], 301 [7d]), *snf4Δ* (622 [1d], 547 [2d], 403 [3d], 440 [4d], 164 [7d]), and *snf1Δ* (555 [1d], 465 [2d], 422 [3d], 434 [4d], 212 [7d]) cultures were grown in glucose-free medium. (C) Quantification of colocalized Rpn2-mC and Hsp42-GFP in WT (469 cells counted [1d], 402 [2d], 498 [3d], 460 [4d], 435 [7d]), *snf4Δ* (355 [1d], 573 [2d], 704 [3d], 494 [4d], 499 [7d]), and *snf1Δ* (459 [1d], 445 [2d], 555 [3d], 535 [4d], 348 [7d]) live cells in 0.025% glucose. (D) Percentage of living cells with colocalized Rpn2-mC and Hsp42-GFP. WT (347 cells counted [1d], 481 [2d],

601 [3d], 396 [4d], 367 [7d]), *snf4Δ* (471 [1d], 541 [2d], 352 [3d], 385 [4d], 183 [7d]), and *snf1Δ* (563 [1d], 415 [2d], 325 [3d], 347 [4d], 138 [7d]) cultures were grown in glucose-free medium. Results plotted as mean±sd..
(TIF)

**S4 Fig. Reversible PSGs are assembled in macroautophagy mutant and vacuolar protease-deficient mutant cells.** (A) Epifluorescence images of Pre10-GFP, Rpn5-GFP, and Rpn2-GFP in low glucose-starved core macroautophagy mutants (*atg11Δ*, *atg17Δ*, *atg39Δ*, *atg15Δ*) from figure panel (5A). (B) Epifluorescence images of core macroautophagy mutants (*atg8Δ*, *atg11Δ*, *atg17Δ*, *atg39Δ*, *atg15Δ*) and vacuolar protease-deficient mutants (*prb1Δ*, *pep4Δ*, *pep4Δ prb1Δ*) cells at the indicated time recovery in 2% glucose. Cells were from figure panel (5A). White arrows mark PSGs in panel (A) and the nucleus in panel (B). (C) Anti-GFP immunoblot analyses of WT and a vacuolar lipase-deficient mutant *atg15Δ* cells under nitrogen starvation for ~1 day at 30˚C. (D) Epifluorescence images of nitrogen-starved WT and *atg15Δ* cells from panel (C). White arrowheads mark GFP-tagged full length proteasomes in the vacuole lumen in *atg15Δ* cells. BF: bright field. Scale bars, 5 μm.
(TIF)

**S5 Fig. Catalytically inhibited proteasomes enhance proteasome fragmentation while compromising PSG assembly during glucose starvation.** (A) Anti-GFP immunoblot analyses of Pre6-GFP (a CP subunit, α4), Rpn5-GFP, and Rpn2-GFP in *pdr5Δ* mutant cells. Cells were harvested from cultures in SC medium containing low glucose (0.025% C) containing either DMSO (control) or 50 μM MG132 and grown for ~1 day at 30˚C. (B) Epifluorescence images of control and MG132-treated cells from panel (A). White arrows mark PSGs. BF: bright field. Scale bar, 5 μm.
(TIF)

**S6 Fig. Normal PSG dynamics and proteasome subunit cleavage in *nvj1Δ* cells, which are incapable of piecemeal autophagy of the nucleus.** (A) Anti-GFP immunoblot analyses of Pre10-GFP, Rpn5-GFP, and Rpn2-GFP in *nvj1Δ* mutant cells under low glucose starvation for ~4 days at 30˚C. (B) Epifluorescence images of Pre10-GFP, Rpn5-GFP, and Rpn2-GFP in *nvj1Δ* cells during low glucose starvation and at the indicated times, recovery in 2% glucose. Cells were from figure panel (A). White arrows mark PSGs in the low glucose panels and the nucleus in the glucose refeeding panels. BF: bright field. Scale bar, 5 μm.
(TIF)

**S7 Fig. Cell viability of mutant cells under low glucose conditions.** Cell viability assay of WT cells, ESCRT mutants (*vps27Δ*, *vps28Δ*, *vps25Δ*, *snf7Δ*, *vps4Δ*), AMPK mutants (*snf1Δ* and *snf4Δ*), and proteasome mutants (*sem1Δ* and *pre9Δ*) using propidium iodide (PI) staining at day 4 and day 7 in low glucose (30˚C). Cells counted (n = 4d, 7d) in WT (775, 668), *vps27Δ* (904, 644), *vps28Δ* (706, 652), *vps25Δ* (761, 715), *snf7Δ* (835, 659), *vps4Δ* (636, 654), *snf4Δ* (843, 685), *snf1Δ* (842, 608), *sem1Δ* (713, 731), and *pre9Δ* (676, 742).
(TIF)

**S1 Video. PSGs are associated with the vacuolar membrane invagination in low glucose-starved *atg8Δ pep4Δ prb1Δ* cells.** Confocal time-lapse images of Pre10-GFP showing that CP-containing PSGs were associated with the vacuolar membrane invagination in *atg8Δ pep4Δ prb1Δ* mutant cells that were in low glucose for ~1 day at 30˚C. The time-lapse video was composed of 40 frames of images with 1.27 s scanning time for each frame and played at 4 frames per second; the real time length was 49.47 s for this video.
(MP4)

**S2 Video. PSGs are associated with the vacuolar membrane invagination in low glucose-starved *atg8Δ pep4Δ prb1Δ* cells.** Confocal time-lapse images of Rpn5-GFP showing that lid-containing PSGs were associated with the vacuolar membrane invagination in *atg8Δ pep4Δ prb1Δ* mutant cells that were in low glucose for ~1 day at 30˚C. The time-lapse video was composed of 40 frames of images with 1.27 s scanning time for each frame and played at 4 frames per second; the real time length was 49.47 s for this video.
(MP4)

**S3 Video. PSGs are associated with the vacuolar membrane invagination in low glucose-starved *atg8Δ pep4Δ prb1Δ* cells.** Confocal time-lapse images of Rpn2-GFP showing that base-containing PSGs were associated with the vacuolar membrane invagination in *atg8Δ pep4Δ prb1Δ* mutant cells that were in low glucose for ~1 day at 30˚C. The time-lapse video was composed of 40 frames of images with 1.27 s scanning time for each frame and played at 4 frames per second; the real time length was 49.47 s for this video.
(MP4)

**S1 Table. Lists of hits from genetic screening of yeast deletion library.**
(XLSX)

**S2 Table. Yeast strains used in this study.**
(DOCX)

## Acknowledgments

We thank Christopher Hickey, Lauren Budenholzer, Chin Leng Cheng, and Mengwen Zhang for critical reading of the manuscript. We thank Shay Ben-Aroya for sharing the *RPN5-GFP-*integrated yeast deletion library for our screen.

## Author Contributions

**Conceptualization:** Jianhui Li, Mark Hochstrasser.

**Formal analysis:** Jianhui Li.

**Funding acquisition:** Maya Schuldiner, Mark Hochstrasser.

**Investigation:** Jianhui Li, Michal Breker, Morven Graham.

**Project administration:** Mark Hochstrasser.

**Resources:** Maya Schuldiner, Mark Hochstrasser.

**Supervision:** Mark Hochstrasser.

**Validation:** Jianhui Li.

**Visualization:** Jianhui Li.

**Writing – original draft:** Jianhui Li.

**Writing – review & editing:** Jianhui Li, Maya Schuldiner, Mark Hochstrasser.

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
