## [Decision Letter · Decision Letter 0]

24 Sep 2019

Dear Dr. Hochstrasser,

Thank you very much for submitting your Research Article entitled 'AMPK regulates ESCRT-dependent microautophagy of proteasomes concomitant with proteasome storage granule assembly during glucose starvation' to PLOS Genetics. Your manuscript was evaluated at the editorial level and by three independent peer reviewers. Each of the reviewers highlighted the novelty and importance of your discoveries; moreover, in general, the reviewers concluded that your studies were well conducted and the majority of the data were convincing. However, as you will see in the detailed evaluations, each of the reviewers believes that revisions are required prior to acceptance. Numerous of the comments could be addressed by altering the text for clarification or to address previous data. However, other comments will require providing new information/experiments such as more details on the results of your screen, additional support data and/or controls,  and quantitation of data presented in images. 

We therefore ask you to modify the manuscript according to the review recommendations. Your revisions should address the specific points made by each reviewer.

[LINK]

Thank you for submitting this interesting and important story to PLoS Genetics. Please let us know if you have any questions while making these revisions.

Yours sincerely,

Anita K. Hopper

Associate Editor

PLOS Genetics

Gregory P. Copenhaver

Editor-in-Chief

PLOS Genetics

Reviewer's Responses to Questions

**Comments to the Authors:**

Reviewer #1: PGENETICS-D-19-01412

The work describes an AMPK Snf1- and ESCRT-dependent sorting pathway for micro-autophagy of presumably aberrant proteasomes versus the sequestration of still functional proteasomes into proteasome storage granules (PSGs) during glucose starvation. The work is based on Yeast Cell Biology by using in vivo fluorescence microscopy of yeast cells expressing GFP-labelled proteasomal subunits, electron microscopy and Western blot analysis.

Despite limitations of these techniques the work is interesting and will inspire the scientific community to analyze quality control pathways not only of proteasomes but also of other giant protein complexes which might be recycled or stored in liquid liquid phase condensates depending on AMPK Snf1.

Several points must be addressed before a revision of the paper can be considered.

Major points:

Fig. 1E, immunogold labelling was performed with anti-CP (Enzo #BML-PW9355),

Please show a Western blot of total lysate and purified proteasomes for antibody specificity. EM images do not show gold particles accumulating in a spherical ‘organelle’ with ~ 500 nm diameter. Achievements over previous EM images showing clear clusters of gold particles with ~ 100-200 nm diameter must be discussed (Laporte et al. 2008, JCB). Did GFP antibodies result in similar immuno gold proteasome localization ?

Fig. 6E. One to two gold particles on an EM micrograph are not statistically relevant for making sweeping claims. Please provide quantification for justification.

p.11, 232 Where do proteasomes localize in vps4 snf1 double null mutants ?

p. 11, 228 What is the evidence that Vph1-GFP is suited as a marker for the class E-like compartment ? Would Cps1 be a better choice to identify this dot at the vacuole as a class-E like compartment (jcs.biologists.org/content/125/21/5208) ? Fig. 3B requires a statistical analysis for the coincidence of Vph1 with proteasomes in the same spot at the vacuole. Moreover, V-type ATPase (Vph1 subunit) disassembles with glucose deprivation (Martinez and Munoz, 2008). Is Vph1 then invaginated into the vacuole at the class E-like compartment for lysosomal degradation ? The rational for choosing Vph1 as class E marker is unclear.

p. 16, 342-345. To create cells bearing large numbers of misassembled or abnormal proteasomes…how is this possible, since after 4 days growth to stationary phase the new synthesis of proteasomal precursor complexes is stalled. Will all proteasomes in sem1 null and pre9 null be dysfunctional, thus not sequestered into PSGs ? Did you also analyze ump1 null mutants with severe defects in proteasome assembly ? They show giant PSG formation. Why are aberrant proteasomes in sem1, pre9 and ump1 null not located in IPODs, if only functional proteasomes are worth to be sequestered into PSGs ?

p. 50, 1067 What is an irreversible PSG ? Per definition PSGs are reversible and motile. Are irreversible PSGs converted into IPOD ?

p. 20, 452 ‘adjacent to the vacuolar membrane during PSG assembly’ is misleading…Laporte et al. 2008 (Fig. 2B) clearly showed that PSGs arise at the NE, when proteasomes exit the nucleus during the transition from proliferation to quiescence.

Figs. 1 to 3 Fluorescence microscopy studies need to be quantified, n≥200 cells, also for supplementary Figures.

Fig. 2B. The vacuolar membrane can hardly be seen by non-experts. The 4X magnifications are dark grey pixels on a light grey background with no clue how the vacuolar membrane was assigned by the doted white line. The vacuolar marker FM4-64 or MDY-64 should be used to visualize the vacuolar membrane in red or yellow, while proteasome dots are green.

Figs. 2A, 3A and 5A. Are the Western blots representative for three biological replicates?

The proteasome dots at the vacuole seem to be close to the nucleus. Did you also check for piecemeal microphagy by using Nvj1 as marker ?

Fig. S5. According to Marshall & Vierstra the addition of MG132 for 4 h induces proteasome dots in yeast (eLife 2018). Kaganovich et al. (2008 Nature) observed JUNQ upon MG132 exposure. JUNQ and PSGs are both reversible and may describe the same membraneless organelle. How do the authors explain the discrepancy between these observations ? Was MG132 after 1 day treatment still effective ?

Minor points:

p. 3, 55; p.4, 80 What is the evidence for phase separation of PSGs ?

It should be mentioned that Snf1 was also identified in a previous high-content screen for PSG formation, Gu et al. 2017 Fig. 1B.

p.10, 222 S2A Fig. turn around

p. 12, 262-265 unclear longwinded sentence

p. 14, 297-299 Is the vacuole acidification dependent of the cytoplasmic pH and the amount of glucose in the cytoplasm ? If the vacuolar pH optimum is not reached, the vacuolar proteases may not be fully active leading the proteasome fragmentation. Please comment.

p. 19, 413 as true for blm10 null mutants (Weberruss et al. 2013)

p. 21 last paragraph about LDs is distracting and can be deleted

p. 33, 793-794 repetition of sentence before

p. 38, exact number of cells can be replaced by n≥200

In the discussion I miss a clear statement about the authors’ new findings with regard to Ben-Aroya’s paper (ref. 17) who already proposed a sorting mechanism of PSGs versus IPODs at the endosome. Own achievements will not be diminished, if previous work from other groups is honestly acknowledged. There are only a couple of papers on PSG which we should know by heart and cite properly.

Reviewer #2: In this manuscript, Li et al. study the dynamics of proteasome assemblies and proteasome quality control in the face of various nutrient stresses, and the contribution of yeast AMPK and ESCRT complexes in this regulation.

Summary and general assessment of the manuscript

The authors use a high-content imaging screening approach to look at proteasome storage granules (PSG) dissipation upon glucose refeeding. The fact that both AMPK, a well-known energy-sensing system, and ESCRT proteins, known to be involved in the protein targeting to the vacuole through various processes such as MVB sorting, microautophagy, etc, are identified is very interesting and indeed merits further consideration.

Then, the authors move on to the study of proteasome macroautophagy induced under nitrogen limitation. The authors describe that ESCRT proteins are required for this process, and as such, this is reminiscent of ESCRT-dependent microautophagy described in the same conditions (Ref 22). Proteasome subunits localize to perivacuolar dots in ESCRT mutants but whether they are of endosomal origin is not addressed; similarly, the mechanism by which ESCRT may help proteasome degradation remains unexplored at this stage.

Building upon previous work by the Ohsumi lab (Ref 41), the authors draw similarities between proteasome degradation and autophagy induction during C starvation. Similar to what has been described, they observe that switching cells to low glucose (0.025% in their case) triggers proteasome degradation, whereas this is barely the case for the no-glucose condition (in which there are no major changes in the abundance of FL proteins, except maybe for Rpn2). Again, proteasome degradation requires most of the ESCRT components tested as observed under N starvation, but this time it also involves AMPK, in line with the fact that autophagy induction in these conditions is AMPK dependent (Ref 41).

It should also be noted that this degradation is not complete - only partial degradation occurs, and the amount of free GFP is overall comparable in the WT and the mutants tested. However, there are differences in the proteolytic fragments likely resulting from incomplete digestion, which the authors use a readout. This partial digestion, or as the authors put it, this “fragmentation”, appears to occur before internalization in the vacuole since proteasome subunits localize at the periphery of the vacuolar membrane in the aberrant “class E” compartments observed in ESCRT mutants. As shown later in the manuscript, this fragmentation is independent of many autophagy genes, but still required vacuolar proteases, which is a bit puzzling.

Another perivacuolar compartment that is important for PSG biology is the protein deposit compartment, IPOD as defined by the marker Hsp42. Co-localization analyses show a strikingly increased association of Pre1 with the IPOD in AMPK mutant, suggesting that AMPK normally prevents association of PSG to the IPOD upon switching cells to low glucose conditions. This is an important observation because Snf1 is inhibited by glucose, so I was initially puzzled as to how Snf1 activity could be solicited when glucose is added back. As the authors suggest, AMPK may be required to salvage proteasomes away from IPOD structures or “terminal” PSGs.

A strong point is the author’s efforts to understand the physiological function of these assemblies & associated regulation. The authors hypothesize that aberrant proteasomes may be generated by C stress - a hypothesis that is not further discussed - and that these could be targeted by microautophagy. Genetic or chemical perturbation of proteasome integrity indeed correlate with an alteration of PSG formation and increased proteasome degradation.

Overall, I found the topic very interesting with a nice set of supporting observations which are technically very well performed. A limitation is that the study is quite broad, making interesting connections between PSG dynamics, quality control and proteasome degradation, AMPK signaling, ESCRT biology, N- starvation and C-starvation or even C removal, autophagy, etc. and consequently many questions remain unanswered. In the end, key questions such how AMPK controls PSG dissipation, how ESCRT “collect” proteasomes for their degradation and why all ESCRT subunits are not always required, how some proteasome subunits escape this regulation, to what do proteasome “fragmentation” and “full degradation” correspond, remain unanswered. Most of these questions are mentioned in the discussion with possible hypotheses, and I understand that each of these questions could be a study in itself, but I was left under the impression that the manuscript is a collection of many observations and is sometimes too superficial.

I am also not sure that the order of appearance of the findings in the manuscript is optimal. The reader is carried from Glc refeeding to N starvation, then to C starvation, then back to Glc refeeding, then back to autophagy, etc. and I think that shuffling some parts could generate a more readable manuscript. Of course, this is just my opinion, but maybe the authors can give it a thought.

Specific comments.

-Fig 1, B and D: the use of profiles from one cell is questionable and doesn’t add much. These profiles can vary a lot depending on how they are drawn across the nucleus and in the end, I think the picture is more informative. Same for S1, B.

-p7 l.137: From the results section, it is not clear how the authors proceed to “refeed” cells with glucose and see PSG dissipation - this could be better explained.

-The description of the results of the high-content imaging screen is vague - the precise number of hits is not disclosed, neither is the list. The authors mention “enrichment” but no values are given.

-It came as a surprise that ESCRT proteins display defects in PSG formation, a finding that was not reported in the previous paper (Ref 16) which identified genes involved in PSG formation, also using Rpn5 as a proxy. Is there something I didn’t get, or could the authors elaborate?

- p8, l.175: the references cited concern mammalian cells, very few studies link AMPK with autophagy in yeast.

- Fig. 3B: Do proteasome subunits accumulate in the class E compartment in vps37∆ mutant? If yes, then how could Vps37 be dispensable?

- Fig 6D - why is it that the proteasome subunits are now in the vacuolar lumen in the vps4∆ background, whereas they were perivacuolar by fluorescence microscopy (3B). How representative are these images?

- I am wondering how the authors can be sure that the proteolytic fragments observed are degraded in vivo and not during protein preparation. The authors use alkaline lysis, which may not be as stringent regarding endogenous protease activity as is cell lysis in TCA. Have the authors considered this?

Reviewer #3: In this manuscript, Li et al. explore the mechanism of proteasome degradation during glucose starvation in yeast. In a genome-wide screen for deletion mutants defective in this process they identified AMPK and ESCRT. The effect of these mutations on the dynamics of PSGs is determined here using microscopy and immune-blot analyses of proteasomal proteins. Based on their results the authors draw a number of major decisions. First, under nitrogen starvation ESCRT, but not AMPK, plays a role in macro-autophagy of the proteasome. In contrast, ESCRT and AMPK mediate the assembly under glucose depletion and disassembly when glucose is added back, respectively, of PSGs and their delivery to the lysosome through micro-autophagy. Second, mostly aberrant proteasomes are delivered to the vacuole via micro-autophagy. Third, without AMPK, CT, but not RP, are sequestered into iPOD, showing different regulation of the two parts of proteasomes.

It was previously shown that while macro-autophagy is important during nitrogen starvation, micro-autophagy is important during glucose starvation. It is also known that ESCRT plays a role in both macro- and micro-autophagy. Evidence presented here shows that this is true also for proteasomes. Because proteasome homeostasis is crucial for cells, this is valuable information. The really novel conclusions are identification of AMPK as a regulator of micro-autophagy and of iPOD as an alternative to PSGs for CP. Showing preferential delivery of aberrant proteasomes for degradation is also important. I have a number of criticisms and suggestions that I think would improve the manuscript

Major criticism:

1. Evidence for micro-autophagy is shown in Figure 6. The data needs quantification and a statement of the number of repeats (this is true for a number of other results, especially figure 7). The microscopy and EM analyses need controls of cells grown with glucose and under nitrogen starvation, and most importantly, to show AMPK mutant control. Finally, I suggest using the mutants with aberrant proteasomes (Figure 7) in this experiment.

2. “Proteasome fragmentation” is partial degradation of proteasomal proteins. The authors need to discuss why would the way proteasomes get to the vacuole would determine whether they would be fully or partially degraded. Is it true also for other cargos?

3. Figure 1: PSGs are identified as fluorescent blobs of a single tagged marker. I think that at least two markers should be used together.

4. Figure 2: The statement in the title (page 8) that ESCRT is required for macro-autophagy is not justified. A comparison between atgX∆ and ESCRT mutants would show whether it is required or plays a role.

5. Figure 2: How do the authors explain that CP is not delivered in the vacuole in vps4∆, while it is delivered in other ESCRT mutants?

6. Figure 3 is confusing: A. Under glucose starvation free GFP is seen in all mutants. Why? ESCRT mutants look like WT: why the authors conclude that ESCRT is involved? B. Why PSG co-localize with compartment E?

Other points:

1. It would be good to show that the mutant cells used here are alive after 4-7 days in glucose deprivation.

2. Clarify if PSGs form also under nitrogen starvation

3. Page 18, line 6: “probably facilitating sealing of … autophagosomes”: Shown in Zhou et al., JCB 2015.

4. Specify throughout the text and legends whether proteasome subunits used are CP or RP.

**Have all data underlying the figures and results presented in the manuscript been provided?**

Reviewer #1: Yes

Reviewer #2: Yes

Reviewer #3: Yes

PLOS authors have the option to publish the peer review history of their article (what does this mean?). If published, this will include your full peer review and any attached files.

Reviewer #1: No

Reviewer #2: No

Reviewer #3: No

---

## [Decision Letter · Decision Letter 1]

7 Nov 2019

Dear Dr Hochstrasser,

We are pleased to inform you that your manuscript entitled "AMPK regulates ESCRT-dependent microautophagy of proteasomes concomitant with proteasome storage granule assembly during glucose starvation" has been editorially accepted for publication in PLOS Genetics. Congratulations!

[GPC note: Reviewer #2 had some final comments (see below) that you may want to consider as you prepare you final draft for the production team (the editorial team will not need to reevaluate).]

Thank you for submitting this interesting study to PLOS Genetics and for supporting open-access publishing; we are looking forward to publishing your work in PLOS Genetics!

Yours sincerely,

Anita K. Hopper

Associate Editor

PLOS Genetics

Gregory P. Copenhaver

Editor-in-Chief

PLOS Genetics

Comments from the reviewers (if applicable):

Reviewer's Responses to Questions

**Comments to the Authors:**

Reviewer #1: The revised version of the manuscript can be accepted, since my major concerns, partly overlapping with reviewer #3, are met. All reviewers agree that the work is interesting and worth to be published.

Still the work is a potpourri of some mysterious observations to be investigated in more detail in future.

No further attachment is uploaded.

Reviewer #2: I thank the authors for considering all of my comments and for their response.

I am overall satisfied with the revision.

It remains that the differential role of ESCRT subunits is not sufficiently discussed for my taste (eg the minor contribution of ESCRT-I subunits such as Vps37 as discussed in the rebuttal p.9/Fig3B) - or did I miss it?. A sentence like “The results suggest that ESCRT-I components do not contribute equally to proteasome degradation and trafficking, perhaps due to different mechanistic roles of ESCRT-I components" (cf rebuttal) would do the job. The data on Vps4 already suggests some differences between MVB sorting and ESCRT-dependent microautophagy. There are other cellular processes where ESCRT function differently than at the MVB (eg. cytokinesis). This could be discussed.

I am also not very much convinced by the data provided concerning proteasome “fragmentation” (rebuttal / p.10). I still don’t understand clearly how this fragmentation happen and this point was also raised by another reviewer.

The additional data provided by the authors show that when doing protein extraction in harsh conditions using TCA, this “fragmentation” no longer occurs. I agree that the rest of the extraction methods seem to show a conserved pattern of degradation and that the TCA seem to give a lower yield of extraction, as pointed out by the authors to justify why they should stick to what they did, but this suggests to me that this degradation occurs in the tube and not in the cell. Of course, there must be some reason why this fragmentation preferentially happens in low glucose vs no glucose or low N but this may have to do with a lot of things (change in global protease activity? Accessibility of proteasome subunits to proteases after breaking the cells etc). Since this only used as a readout to discriminate between various degradation pathway, understanding the basis of this behavior wouldn’t change the conclusions so I’m enclined to suggest the acceptance of the paper despite this.

Reviewer #3: Review PGENETICS-D-19-01412R1

In the revised manuscript and their response, the authors addressed all my concerns.

**Have all data underlying the figures and results presented in the manuscript been provided?**

Reviewer #1: Yes

Reviewer #2: Yes

Reviewer #3: Yes

PLOS authors have the option to publish the peer review history of their article (what does this mean?). If published, this will include your full peer review and any attached files.

Reviewer #1: No

Reviewer #2: No

Reviewer #3: No

**Data Deposition**

http://datadryad.org/submit?journalID=pgenetics&manu=PGENETICS-D-19-01412R1

**Press Queries**

---

## [Editor Report · Acceptance letter]

11 Nov 2019

PGENETICS-D-19-01412R1 

AMPK regulates ESCRT-dependent microautophagy of proteasomes concomitant with proteasome storage granule assembly during glucose starvation 

Dear Dr Hochstrasser, 

We are pleased to inform you that your manuscript entitled "AMPK regulates ESCRT-dependent microautophagy of proteasomes concomitant with proteasome storage granule assembly during glucose starvation" has been formally accepted for publication in PLOS Genetics! Your manuscript is now with our production department and you will be notified of the publication date in due course.

With kind regards,

Nicholas White

PLOS Genetics

On behalf of:
